# A novel mechanism of bulk cytoplasmic transport by cortical dynein in *Drosophila* ovary

**Wen Lu, Margot Lakonishok, Anna S Serpinskaya, Vladimir I Gelfand\***

Department of Cell and Developmental Biology, Feinberg School of Medicine, Northwestern University, Chicago, United States

**Abstract** Cytoplasmic dynein, a major minus-end directed microtubule motor, plays essential roles in eukaryotic cells. *Drosophila* oocyte growth is mainly dependent on the contribution of cytoplasmic contents from the interconnected sister cells, nurse cells. We have previously shown that cytoplasmic dynein is required for *Drosophila* oocyte growth and assumed that it simply transports cargoes along microtubule tracks from nurse cells to the oocyte. Here, we report that instead of transporting individual cargoes along stationary microtubules into the oocyte, cortical dynein actively moves microtubules within nurse cells and from nurse cells to the oocyte via the cytoplasmic bridges, the ring canals. This robust microtubule movement is sufficient to drag even inert cytoplasmic particles through the ring canals to the oocyte. Furthermore, replacing dynein with a minus-end directed plant kinesin linked to the actin cortex is sufficient for transporting organelles and cytoplasm to the oocyte and driving its growth. These experiments show that cortical dynein performs bulk cytoplasmic transport by gliding microtubules along the cell cortex and through the ring canals to the oocyte. We propose that the dynein-driven microtubule flow could serve as a novel mode of fast cytoplasmic transport.

**\*For correspondence:**
vgelfand@northwestern.edu

**Competing interest:** The authors declare that no competing interests exist.

## Editor's evaluation

In their manuscript, Lu et al., use a combination of experimental approaches to determine how cellular components are transported from nurse cells into the growing oocyte during *Drosophila* egg development. The authors demonstrate that the minus-end directed microtubule motor, dynein, generates cortical flow by gliding microtubules along the cell cortex. This action is distinct from dynein's cargo transport functions, as the authors replace dynein with a minus-end directed kinesin linked to the cortex and observe the same phenomenon. This flow is capable of propelling cargoes through the ring canals into the growing oocyte via a bulk cytoplasmic transport mechanism, highlighting a novel mode of fast cytoplasmic transport. Overall, this work had broad significance to cell biologists and developmental biologists.

## Introduction

Microtubules perform many key cellular functions, such as cell division, migration, polarization/compartmentation, and intracellular long-range cargo transport. Cytoplasmic dynein (referred simply as dynein hereafter) is the major minus-end directed microtubule motor, and involved in numerous microtubule-based functions (*Roberts et al., 2013*). In interphase cells, dynein is the main motor responsible for transporting various cargoes towards the microtubule minus-ends (*Reck-Peterson et al., 2018*). In dividing cells, dynein functions at kinetochores, spindle poles, and at the cell cortex.

Particularly, cortical dynein pulls astral microtubules and is therefore required for positioning the mitotic spindles (*Vaughan, 2012*; *Karki and Holzbaur, 1999*).

The core dynein complex contains two copies of heavy chain (DHC), intermediate chain (DIC), intermediate light chain (DLIC) and three different dynein light chains (Roadblock, LC8/Cut up and Tctex/Dlc90F) (*Reck-Peterson et al., 2018*; *Canty and Yildiz, 2020*; *Figure 1A*). Dynein heavy chain contains a ring of six AAA+ domains, and ATP hydrolysis-induced conformational change results in dynein walking towards the minus-ends of microtubules (*Cianfrocco et al., 2015*). The activity of dynein is regulated by the dynactin complex including the largest subunit p150Glued/DCTN1 (*Urnavicius et al., 2015*), and the Lis1-NudE complex (*McKenney et al., 2010*), as well as several activating adaptors, such as BICD2/BICDL1, Spindly, and HOOK1/3 (*Reck-Peterson et al., 2018*; *Canty and Yildiz, 2020*; *Olenick and Holzbaur, 2019*; *Figure 1A*).

Dynein has many essential functions during *Drosophila* oogenesis. First, it is required for germline cell division and oocyte specification (*McGrail and Hays, 1997*; *Liu et al., 1999*). During mid-oogenesis, dynein is required for transport of mRNA ribonucleoproteins (RNPs) and organelles from nurse cells to the oocyte (*Mische et al., 2007*; *Clark et al., 2007*; *Nicolas et al., 2009*; *Lu et al., 2021*). Within the oocyte, dynein transports and anchors the anterior and dorsal determinants that are critical for axis determination for future embryos (*Januschke et al., 2002*; *Duncan and Warrior, 2002*; *Trovisco, 2016*). During vitellogenesis, dynein in the oocyte regulates endocytic uptake and maturation of yolk proteins from the neighboring somatic follicle cells (*Liu et al., 2015*).

The *Drosophila* oocyte undergoes dramatic cell growth and polarization during oogenesis (*Bastock and St Johnston, 2008*). Remarkably, the oocyte remains transcriptionally quiescent during most of the oogenesis. For its dramatic growth, the oocyte relies on its interconnected sister cells, nurse cells, for providing mRNAs, proteins, and organelles through intercellular cytoplasmic bridges called ring canals (*Bastock and St Johnston, 2008*; *Mahajan-Miklos and Cooley, 1994*). Previously, we showed that dynein heavy chain drives oocyte growth by supplying components to the growing oocyte (*Lu et al., 2021*). Here, we study the mechanism of dynein-driven transport of cargoes from nurse cells to the oocyte. We find that microtubules, which had previously been considered as static tracks for dynein (*Mische et al., 2007*; *Clark et al., 2007*; *Nicolas et al., 2009*; *Lu et al., 2021*), are robustly moved by dynein within the nurse cell cytoplasm, and more remarkably, from the nurse cell to the oocyte. We further demonstrate that this dynein-powered microtubule gliding creates cytoplasmic advection carrying cargoes in bulk to the oocyte, including neutral particles that do not interact with motors. Furthermore, we show that a chimeric gliding-only minus-end motor anchored to the cortex is sufficient to support transport of cargoes to the oocyte, and therefore to drive the oocyte growth. Finally, we describe here a novel mechanism for dynein-driven cytoplasmic transport: cortically anchored dynein drives microtubule gliding, and microtubules in turn move cytoplasmic contents in nurse cells and through the ring canals, driving oocyte growth. This provides a fast and efficient mode of bulk transport of a wide variety of components supplied by nurse cells to the oocyte.

## Results

### Dynein is required for oocyte growth

The growing *Drosophila* oocyte is transcriptionally silent and mostly relies on its interconnected sister nurse cells for mRNAs, proteins, and organelles (*Bastock and St Johnston, 2008*; *Mahajan-Miklos and Cooley, 1994*). Dynein, the main minus-end directed microtubule motor in *Drosophila*, has been implicated in nurse cell-to-oocyte transport (*Mische et al., 2007*; *Clark et al., 2007*; *Nicolas et al., 2009*; *Lu et al., 2021*).

Here, we investigated the roles of the dynein complex and its regulators in oocyte growth by taking advantage of a germline-specific Gal4, maternal α tubulin-Gal4[V37], that is expressed in germline cells after the completion of cell division and oocyte specification (*Lu et al., 2021*; *Sanghavi et al., 2016*). This approach bypasses the requirement for dynein in the germarium. Knockdown of dynein by expressing either RNAi against dynein components (DHC, DLIC, Lis1) or a dominant negative construct of p150Glued/DCTN1 (*p150GluedΔC*) (*Mische et al., 2007*) driven by this Gal4 line allowed for normal cell division and oocyte specification but caused complete arrest of oocyte growth (hereafter referred as the 'small oocyte' phenotype) (*Figure 1B–D"*; *Figure 1—figure supplement 1A*). In spite of the growth inhibition caused by dynein knockdown, we found that the oocyte marker, Orb (oo18



**Figure 1.** Dynein activity is required for *Drosophila* oocyte growth. (**A**) A cartoon illustration of cytoplasmic dynein and its regulators. The dynein core complex is composed of dimers of dynein heavy chain (orange), dynein intermediate chain (gray), dynein light intermediate chain (blue), and three types of dynein light chains (green). Dynein activity is regulated by the dynactin complex (brown, with p150Glued highlighted in red) and the Lis1-NudE complex (Lis1, yellow; NudE, magenta). A dynein activating adaptor, BicD (purple), is also shown to illustrate the linkage of the dynein complex with a cargo. To

*Figure 1 continued on next page*

*Figure 1 continued*

note: other cargo adaptors instead of BicD, such as Spindly, HOOK1/3, ninein/ninein- like(NINL), and RAB11 family-interacting protein 3, can be used for dynein activation and cargo recruitment (not shown) (*Reck-Peterson et al., 2018*). BICDR1 and HOOK3 could recruit two dyneins for increased force and speed (not shown) (*Reck-Peterson et al., 2018*). (**B**) Summary of oocyte growth phenotypes in listed genetic background (all with one copy of *maternal αtub-Gal4*[V37]). Classifications of oocyte growth phenotypes are included in *Figure 1—figure supplement 1A*. *Dhc64C-RNAi* (#1) is the RNAi line used for all *Dhc64C-RNAi* experiments in this study. (**C-D"**) Phalloidin and Orb staining in control (**C-C"**) and *Dhc64C-RNAi* (**D-D"**) ovarioles. Oocytes and Orb staining are highlighted with either yellow arrowheads and brackets (**C-D and C"-D"**), or with yellow painting (**C'-D'**). Scale bars, 50 µm. (**E**) Summary of the Orb staining phenotypes in stage 8 (left) and stage 9 (right) egg chambers in listed genotypes (all with one copy of *maternal αtub-Gal4*[V37]). Descriptions of Orb concentration and Orb dispersion are included in *Figure 1—figure supplement 1B*. (**F**) Summary of oocyte growth phenotypes in *RNAi* lines against three listed dynein activating adaptors, BicD, Spindly and Hook (all with one copy of *maternal αtub-Gal4*[V37]).

The online version of this article includes the following figure supplement(s) for figure 1:

**Figure supplement 1.** Summary of dynein-related RNAi phenotypes.

RNA-binding protein) (*Lantz et al., 1994*) is properly concentrated in the early oocytes, emphasizing that our approach does not interfere with germline cell division or oocyte specification. However, Orb is clearly dispersed from the small oocytes at later stages (stages 8–9) (*Figure 1C", D", E*; *Figure 1—figure supplement 1B*), implying defects of nurse cell-to-oocyte transport. These data demonstrate that dynein core components and regulators are indeed essential for oocyte growth, likely via transporting cargoes into the oocyte.

As dynein activity relies on various cargo activating adaptors, we knocked down the *Drosophila* homologs of three main dynein cargo-specific adaptors, BicD, Spindly, and Hook (*Olenick and Holzbaur, 2019*; *Lee et al., 2018*) by RNAi in the germ line. Among the three adaptors, knockdown of BicD inhibits the oocyte growth, while *Spindly-RNAi* and *hook-RNAi* do not cause obvious oocyte growth defects (*Figure 1F*).

This lack of oocyte growth phenotype in *Spindly-RNAi* and *hook-RNAi* animals is not due to low efficiency of the RNAi lines themselves, as knockdown of Spindly and Hook using these RNAi lines display typical loss-of-function mutant phenotypes of these adaptors. Maternal knockdown of Spindly (*mat αtub*[V37]> *Spindly-RNAi*) caused all embryos to fail to hatch (N > 200) and zygotic knockdown using a strong ubiquitous Gal4 (*Actin5C-Gal4*) led to 0% eclosion rate from *Spindly-RNAi* pupae (*Figure 1—figure supplement 1C*; *Clemente et al., 2018*). Hook is not required for fly viability or fertility but is required for proper bristle formation (*Krämer and Phistry, 1999*), and *hook-RNAi* animals phenocopied the classic hooked-bristle phenotype observed in *hook* null allele (*hook*[11]) (*Figure 1—figure supplement 1D–F'*; *Krämer and Phistry, 1999*).

This set of data leads us to conclude that BicD is the most important dynein activating adaptor for *Drosophila* oocyte growth.

## Dynein drives microtubule gliding in nurse cells

Having established that dynein and its associated proteins are required for the oocyte growth, we next examined dynein tracks, cytoplasmic microtubules. Microtubules are localized inside the intercellular cytoplasmic bridges, the ring canals (*Figure 2—figure supplement 1A-A""*), consistent with previous reports (*Mische et al., 2007*; *Clark et al., 2007*; *Nicolas et al., 2009*; *Lu et al., 2021*). However, when we examined the dynamics of microtubules in live samples using photoconversion (*Lu et al., 2013b*; *Lu et al., 2015*; *Lu et al., 2016*), we discovered that microtubules in the nurse cells are not stationary; they vigorously move and snake around in the nurse cells (*Figure 2A–A'*; *Figure 2—video 1*). Robust microtubule movement in nurse cells can also be seen with fluorescently labeled microtubule-associated proteins (MAPs), Ensconsin/MAP7 (EMTB) (*Lu et al., 2020*; *Faire et al., 1999*), Patronin/CAMSAP (*Lu et al., 2021*; *Lu et al., 2020*), and Jupiter (*Lu et al., 2013b*; *Lu et al., 2015*; *Lu et al., 2020*; *Karpova et al., 2006*; *Figure 2—videos 2–4*) as early as in stage 6 egg chambers. Even more remarkably, microtubules are seen moving from the nurse cell to the oocyte through the ring canal. This movement is most robust in stage 9 egg chambers and has an average velocity of ~140 nm/s (*Figure 2C-E, H*; *Figure 2—videos 1–4*). Thus, microtubules are not just static tracks for dynein; instead, they actively move within the nurse cells and from the nurse cells to the oocyte.

When the *Drosophila* egg chambers reach stages 10B-11, nurse cells transfer all their cytoplasmic contents to the oocytes, the process called nurse cell dumping (*Mahajan-Miklos and Cooley, 1994*; *Buszczak and Cooley, 2000*). We examined whether the microtubule movement in the ring canals



**Figure 2.** Microtubule movement within nurse cells and from nurse cells to the oocyte. (**A-B′**) Microtubule movement labeled with photoconverted tdMaple3-αtub in control and *Dhc64C-RNAi* nurse cells. Photoconversion area is highlighted with a dotted purple circle and microtubules outside of the photoconversion zone are highlighted with purple arrowheads. Scale bars, 20 μm. See also *Figure 2—video 1* and *Figure 2—video 5*. (**C-C″**) Microtubule movement is visualized by a 3XTagRFP-tagged microtubule binding domain of human Ensconsin/MAP7 (EMTB-3XTagRFP). Temporal

*Figure 2 continued on next page*

*Figure 2 continued*

color-coded hyperstacks are used to show the microtubule movement in a whole egg chamber (**C**) and zoom-in areas (the dashed orange boxes) within a nurse cell (**C'**) and in a nurse cell-oocyte ring canal (**C"**). Scale bar, 50 µm. See also *Figure 2—video 2*. (**D-D'''**) microtubule movement labeled with a GFP-tagged microtubule minus-end binding protein Patronin. The ring canal is labeled with LifeAct-TagRFP. One microtubule moving through the ring canal is highlighted with orange arrowheads. Scale bars, 10 µm. See also *Figure 2—video 3*. (**E**) Microtubule movement is visualized by a GFP protein trap line of an endogenous microtubule binding protein, Jupiter (Jupiter-GFP). The ring canal is labeled with F-Tractin-tdTomato. A kymograph of Jupiter-GFP in the nurse cell-oocyte ring canal (the dashed orange box) is used to show the microtubule movement from the nurse cell to the oocyte. See also *Figure 2—video 4*. (**F**) Cytoplasmic advection is visualized by bright-field imaging. The ring canals are labeled with a GFP-tagged myosin-II light chain, Sqh-GFP. Kymographs of the two nurse cell-oocyte ring canals (the dashed white boxes) are used to show the cytoplasmic advection from the nurse cells to the oocyte. See also *Figure 2—video 7*. (**G**) Bulk movement of two types of cargoes, mitochondria (Mito-MoxMaple3, without photoconversion, magenta) and Golgi units (RFP-Golgi, cyan), through the nurse cell-oocyte ring canal, labeled with GFP-Pav (magenta). Kymographs of mitochondria and Golgi units in the nurse cell-oocyte ring canal (the dashed orange box) is used to show that both cargoes move at a similar speed through the ring canal to the oocyte. See also *Figure 2—video 8*. (**E–G**) The capped line on top of the kymograph indicates the ring canal region. Scale bars, 50 µm. (**H**) Quantification of the velocities of microtubules, small particles in bright-field images (BF particles) and mitochondria in nurse cell-oocyte ring canals of stage 9 egg chambers. The black bars on top of scattered plots stand for mean ± 95% confidence intervals: microtubules (Jupiter-GFP) 139.7 ± 6.2 nm/s (N = 159); BF particles, 140.8 ± 6.4 nm/s (N = 143); mitochondria (Mito-MoxMaple3), 117.6 ± 5.7 nm/s (N = 147). Unpaired t tests with Welch's correction were performed in following groups: microtubules and BF particles, p = 0.8080 (n.s.); microtubules and mitochondria, p < 0.0001 (****); BF particles and mitochondria, p < 0.0001 (****). (**I**) Measurement of oocyte sizes in egg chambers of different stages (mean ± 95% confidence intervals): stage 4, 67.2 ± 4.9 µm² (N = 57); stage 5, 139.0 ± 12.6 µm² (N = 58); stage 6, 230.5 ± 13.4 µm² (N = 62); stage 7, 378.4 ± 25.1 µm² (N = 38); stage 8, 1067.3 ± 120.0 µm² (N = 41); stage 9, 5432.8 ± 771.8 µm² (N = 69); stage 10 A, 20105.0 ± 1930.0 µm² (N = 31).

The online version of this article includes the following video and figure supplement(s) for figure 2:

**Figure supplement 1.** Microtubule in the ring canals and oocyte growth at different stages.

**Figure supplement 2.** Myosin-II activity is dispensable for oocyte growth during mid-oogenesis.

**Figure 2—video 1.** Microtubule movement (labeled with locally photoconverted tdMaple3-αtub) in nurse cells and from the nurse cell to the oocyte via a ring canal (labeled with *ubi-GFP-Pav*).

https://elifesciences.org/articles/75538/figures#fig2video1

**Figure 2—video 2.** Microtubules labeled with fluorescently tagged human MAP7/Ensconsin microtubule binding domain (EMTB) move in nurse cells and through the nurse cell-oocyte ring canal in stage 9 egg chambers.

https://elifesciences.org/articles/75538/figures#fig2video2

**Figure 2—video 3.** Microtubules labeled with an ectopically-expressed GFP-tagged minus-end binding protein, Patronin (GFP-Patronin) move in nurse cells and from the nurse cell to the oocyte through the ring canal (labeled with LifeAct-TagRFP in magenta).

https://elifesciences.org/articles/75538/figures#fig2video3

**Figure 2—video 4.** Microtubules labeled with a GFP-tagged endogenous microtubule-associated protein Jupiter (Jupiter-GFP) in nurse cells and in the nurse cell-oocyte ring canal (labeled with F-Tractin-tdTomato in magenta).

https://elifesciences.org/articles/75538/figures#fig2video4

**Figure 2—video 5.** No microtubule movement (labeled with locally photoconverted tdMaple3-αtub) in *Dhc64C-RNAi* nurse cells, compared to control.

https://elifesciences.org/articles/75538/figures#fig2video5

**Figure 2—video 6.** Microtubule movement (labeled with a GFP-tagged endogenous MAP, Jupiter-GFP) in nurse cell-oocyte ring canals (labeled with GFP-Pav) in control, *zip-RNAi* and *Dhc64C-RNAi*.

https://elifesciences.org/articles/75538/figures#fig2video6

**Figure 2—video 7.** Cytoplasmic advection from the nurse cells to the oocyte through ring canals (labeled with Sqh-GFP in magenta).

https://elifesciences.org/articles/75538/figures#fig2video7

**Figure 2—video 8.** Bulk movement of mitochondria (labeled with Mito-MoxMaple3, without photoconversion, magenta) and Golgi units (labeled with RFP-Golgi, cyan) through the nurse cell-oocyte ring canal (labeled with GFP-Pav, magenta) in a stage 9 egg chamber.

https://elifesciences.org/articles/75538/figures#fig2video8

**Figure 2—video 9.** Mitochondria movement (labeled with Mito-MoxMaple3, after global photoconversion, gray) in control, *zip-RNAi* and *Dhc64C-RNAi*.

https://elifesciences.org/articles/75538/figures#fig2video9

and associated oocyte growth during mid-oogenesis is a result of an early slow form of nurse cell dumping. We first measured the nurse cell size between stage 8 to stage 10 and found that nurse cells still undergo dramatic growth during this phase (*Figure 2—figure supplement 2A*). This nurse cell growth from stage 8 to stage 10 is quite distinct from the dumping phase, when nurse cells squeeze their cytoplasm to the oocyte and thus quickly shrink (*Gutzeit, 1986*).

Second, nurse cell dumping requires the activity of non-muscle myosin-II (*Wheatley et al., 1995*; *Jordan and Karess, 1997*). We tested whether myosin-II is required for stage 9 microtubule movement and overall oocyte growth. We used a RNAi line against the myosin-II heavy chain Zipper (*zip-RNAi*) to knockdown myosin-II activity, and an antibody recognizing phosphorylated myosin-II light chain (p-MLC, Ser19) as a readout of myosin-II activity (*Jordan and Karess, 1997*; *Sun et al., 2011*). We found that that myosin-II activity is dramatically diminished in *zip-RNAi* egg chambers (*Figure 2—figure supplement 2B–2C'*), suggesting that the *zip-RNAi* does inhibit myosin-II efficiently. However, *zip-RNAi* shows no defects in oocyte growth or Orb concentration from early to mid-oogenesis (*Figure 2—figure supplement 2D–2G*). Additionally, we induced germline clones of a loss-of-function allele of *zip* (*zip²*) (*Halsell et al., 2000*), and found that the majority of *zip²* mutant egg chambers with proper oocyte specification can develop to mid-oogenesis without obvious delay in oocyte growth (25 out of 28 egg chambers; *Figure 2—figure supplement 2H–2K*). This is consistent with previous reports showing that the myosin-II light chain *spaghetti squash* (*sqh*) mutants (*Wheatley et al., 1995*; *Jordan and Karess, 1997*; *Doerflinger et al., 2022*) and the 'dumpless' mutants, such as chickadee, E2F, and RAP150B, develop normally to stages 9~10 without major oocyte growth defects (*Cooley et al., 1992*; *Myster et al., 2000*; *Imran Alsous et al., 2021*).

Dynein is known to glide microtubules in vitro and in vivo (*Lu and Gelfand, 2017*). Therefore, we asked whether dynein is the motor that moves microtubules in nurse cells. Knockdown of dynein using the *Dhc64C-RNAi* line results in a complete inhibition of microtubule movement within the nurse cells (*Figure 2B, B'*; *Figure 2—video 5*). The microtubules in *dynein-RNAi* nurse cells are noticeably less curved than the control ones, implying no motors applying forces to them. More importantly, the motility of microtubules in the nurse cell-oocyte ring canals are dramatically reduced in *dynein-RNAi* (*Figure 2—video 6*). In contrast, we found that the microtubule movement through the ring canal is not affected by myosin-II inhibition (*Figure 2—video 6*). Thus, we conclude that the microtubule movement through the nurse cell-oocyte ring canals is not a part of myosin-II-driven nurse cell dumping; instead, it represents a novel mode of dynein gliding microtubules in nurse cells and through ring canals to the oocyte.

## Microtubule flow carries cargoes to the oocyte

In addition to microtubule motility, we also observed synchronized movement of cargoes through nurse cell-oocyte ring canals in stage 9 egg chambers, at a rate similar to microtubule movement (*Figure 2F–H*; *Figure 2—videos 7; 8*), indicating that microtubule-driven cytoplasmic advection occurs in the ring canals. Furthermore, like microtubule movement, bulk cargo movement through ring canals is dynein-dependent and myosin-II-independent (*Figure 2—video 9*). This bulk cargo movement occurs at stage 9 when the oocyte undergoes the most dramatic growth (*Figure 2I*; *Figure 2—figure supplement 1B, C*). This raises the possibility that, unlike the previously proposed canonical cargo transport (*Figure 3A*), dynein-powered microtubule gliding itself could create cytoplasmic advection carrying all types of cargoes from nurse cells to the oocyte to support the rapid oocyte growth (*Figure 3B*).

Therefore, to test this possibility, we simultaneously imaged microtubules and organelles (mitochondria or Golgi units) in the nurse cell-oocyte ring canals, and found that they move together with microtubules through the ring canal to the oocyte (*Figure 3—videos 1; 2*). This further suggests that mitochondria, Golgi units and probably other classes of cargoes could be carried through ring canals in bulk by microtubules gliding towards the oocyte (*Figure 3B*).

However, we cannot completely exclude the possibility that the motors attached to the cargoes walk on these moving microtubules and thus co-transport cargoes with microtubules through the ring canal. Thus, we decided to examine whether neutral particles that normally are not transported by motors along microtubules can be carried to the oocyte. We used Genetically Encoded Multimeric nanoparticles (GEMs) (*Figure 3C*), which self-assemble into ~40 nm fluorescent spheres (*Delarue et al., 2018*). In *Drosophila* S2R + cells, GEMs form bright compact particles and display mostly Brownian motion, which is very distinct from typical movements of endogenous motor-driven organelles, such as lysosomes (*Figure 3—figure supplement 1*). However, when we expressed GEMs in *Drosophila* egg chambers, we found that they move within nurse cells in a fast linear manner (*Figure 3D–D'*; *Figure 3—video 3*). More importantly, GEMs move through the nurse cell-oocyte ring canals (*Figure 3—video 4*) and concentrate in the oocytes (*Figure 3F–G*). Knockdown of dynein

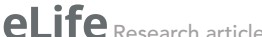

**Figure 3.** Dynein-dependent GEM particle movement in the egg chambers. (**A–B**) Cartoon illustrations of two possible mechanisms of dynein-dependent cargo transfer from the nurse cell to the oocyte. (**C**) A schematic illustration of the GEM construct. 120 copies of a sapphire-tagged *Pyrococcus furiosus* Encapsulin scaffold protein self-assemble into a 40 nm particle. (**D-E′**) Temporal color-coded hyperstacks of GEM particles in control (**D-**

*Figure 3 continued on next page*

*Figure 3 continued*

D') and *Dhc64C-RNAi* (**E-E'**). Oo, the oocyte. Scale bars, 20 μm. See also *Figure 3—video 3*. (**F–G**) Quantifications of total (**F**) and average (**G**) fluorescent intensities of GEM particles in control and *Dhc64C-RNAi*. The values shown in the graphs are mean ± 95% confidence intervals. (**F**) Control, N = 22; *Dhc64C-RNAi*, N = 27. Unpaired t test with Welch's correction between control and *Dhc64C-RNAi*: p < 0.0001 (****). (**G**) Control, N = 22; *Dhc64C-RNAi*, N = 27. Unpaired t test with Welch's correction between control and *Dhc64C-RNAi*: p < 0.0001 (****). (**H–K**) Quantifications of velocities (**H, J**) and trajectories (**I, K**) of GEM movement in control and *Dhc64C-RNAi*. The number of particles tracked: control, N = 7656; *Dhc64C-RNAi*, N = 2083. (**H–I**) The values shown in the graphs are mean ± 95% confidence intervals. (**H**) Unpaired t test with Welch's correction between control and *Dhc64C-RNAi*: p < 0.0001 (****). (**I**) Unpaired t test with Welch's correction between control and *Dhc64C-RNAi*: p < 0.0001 (****). (**J–K**) Histograms of velocities (**J**) and trajectories (**K**) of GEM movement in control and *Dhc64C-RNAi* (the same set of data used in H-I).

The online version of this article includes the following video and figure supplement(s) for figure 3:

**Figure supplement 1.** GEMs and lysosomes in *Drosophila* S2R + cells.

**Figure 3—video 1.** Mitochondria movement (labeled with Mito-MoxMaple3, without photoconversion, left) and microtubule movement (labeled with EMTB-3XTagRFP, right) in a nurse cell-oocyte ring canal (labeled with GFP-Pav, left).

https://elifesciences.org/articles/75538/figures#fig3video1

**Figure 3—video 2.** The movement of Golgi units (labeled with RFP-Golgi, left) and microtubules (labeled with Jupiter-GFP, right) in a nurse cell-oocyte ring canal (labeled with F-Traction-tdTomato, left).

https://elifesciences.org/articles/75538/figures#fig3video2

**Figure 3—video 3.** GEM particles in a control egg chamber and in a *Dhc64C-RNAi* egg chamber.

https://elifesciences.org/articles/75538/figures#fig3video3

**Figure 3—video 4.** GEM particles transport through a nurse cell-oocyte ring canal (labeled with F-Tractin-tdTomato, magenta) in control.

https://elifesciences.org/articles/75538/figures#fig3video4

---

dramatically diminishes GEM linear movements and eliminates GEM accumulation in the oocyte (*Figure 3E–K*; *Figure 3—video 3*). Altogether, we demonstrate that the direct dynein-cargo interaction is not necessary for nurse cell-to-oocyte transport, and neutral particles can be efficiently carried through the ring canals by dynein-driven microtubule movement.

## Microtubule gliding delivers cargoes to oocytes

Next, we investigated whether cortical dynein glides microtubules in nurse cells and transports cargoes to the oocyte. We first examined dynein localization using a GFP-tagged Dlic transgenic line as well as antibodies against *Drosophila* dynein heavy chain and BicD. We found clear cortical localization of dynein in the nurse cells after the soluble pool of dynein is extracted by detergent (*Figure 4—figure supplement 1A–1F*).

Then we tested whether a cortically anchored minus-end-directed motor is sufficient to drive oocyte growth. To constrain dynein activity to cell cortex, we replaced the endogenous dynein activating adaptor BicD with an ectopically expressed BicD that is recruited to the actin cortex by an actin-targeting motif, F-Tractin (*Spracklen et al., 2014*; *Figure 4—figure supplement 1G–1H'*). Compared to *BicD-RNAi* in which no egg chambers developed passing stage 7, this cortically recruited BicD construct allows >70% ovarioles to have egg chambers reaching mid-oogenesis (*Figure 4A–D*). In contrast, expression of a tdTomato-tagged F-Tractin alone (*Spracklen et al., 2014*) did not rescue the oocyte growth defects in *BicD-RNAi* (*Figure 4D*), indicating that the rescue we observed with F-Tractin-BicD is not due to Gal4 activity dilution or F-Tractin overexpression.

To further test the idea that gliding microtubules drive nurse cell-to-oocyte transport, we created an artificial minus-end gliding-only motor. It contains a dimer motor region of a fast minus-end-directed plant kinesin-14, Kin14VIb (*Jonsson et al., 2015*; *Nijenhuis et al., 2020*; *Yamada et al., 2017*) targeted to cell cortex with the F-Tractin probe (*Figure 4—figure supplement 1G*). This chimeric kinesin-14 motor, unlike dynein, cannot carry endogenous *Drosophila* cargoes, allowing us to test whether microtubule gliding alone is sufficient for transporting cargoes to the oocyte and support the oocyte growth. With this chimeric gliding-only kinesin-14 motor, we found that oocyte growth is partially rescued (*Figure 4E–G and K*). In addition to the oocyte size rescue, we also found



**Figure 4.** A cortically anchored minus-end motor is sufficient to drive oocyte growth. (**A–C**) Oocyte growth defect in *BicD-RNAi* is rescued by a cortically restricted BicD construct. (**D**) Summary of percentages of ovarioles with and without egg chamber(s) passing stage 7. Staging of *BicD-RNAi* egg chambers is determined by the nurse cell size. (**E–J**) The defects of oocyte growth and Orb concentration in *Dlic-RNAi* are rescued by a cortically recruited plant Kin14 (Kin14VIb) construct. Phalloidin and DAPI staining (**E–G**) or Orb staining (**H–J**) in control (**E, H**), *Dlic-RNAi* (**F, I**) and *Dlic-RNAi* rescued by F-tractin-GFP-Kin14 (**G, J**). (**K**) Quantification of oocyte size in stage 8~9 egg chambers in listed genotypes. The values shown in the graph are mean ± 95% confidence intervals. Control, N = 40; *Dlic-RNAi*, N = 49; *Dlic-RNAi+ F-Tractin-Kin14*, N = 33; *Dlic-RNAi+ F-Tractin-tdTomato*, N = 42. Unpaired t test with Welch's correction were performed in following groups: between control and *Dlic-RNAi*, p < 0.0001 (****); between control and *Dlic-RNAi+ F-Tractin-Kin14*, p < 0.0001 (****); between control and *Dlic-RNAi+ F-Tractin-tdTomato,* p < 0.0001 (****); between *Dlic-RNAi* and *Dlic-RNAi+ F-Tractin-Kin14*, p < 0.0001 (****); between *Dlic-RNAi* and *Dlic-RNAi+ F-Tractin-tdTomato,* p = 0.2107 (n.s.); between *Dlic-RNAi+ F-Tractin-Kin14* and *Dlic-RNAi+ F-Tractin-tdTomato,* p < 0.0001 (****). (**L**) Summary of Orb staining phenotypes in stage 8 (left) and stage 9 (right) egg chambers in listed genotypes. Oocytes are highlighted with either yellow arrowheads or yellow brackets. All listed genotypes carried one copy of *maternal αtub-Gal4[V37]*. Scale bars, 50 μm.

The online version of this article includes the following video and figure supplement(s) for figure 4:

**Figure supplement 1.** The recruitment of dynein to nurse cell cortex.

**Figure 4—video 1.** Mitochondria movement (labeled with Mito-MoxMaple3, after global photoconversion) in the nurse cell-oocyte ring canals of control, *Dlic-RNAi* and *Dlic-RNAi+ F-Tractin-GFP-Kin14VIb* rescued samples.

https://elifesciences.org/articles/75538/figures#fig4video1

that in more than 95% of the samples, the kinesin-14 motor is able to maintain Orb concentration in the oocyte of stage 8~9 egg chambers (*Figure 4H–J and L*). The rescue of both oocyte growth and oocyte Orb concentration implies that the gliding-only motor restores the nurse cell-to-oocyte transport.

We directly examined mitochondria movement in the kinesin-14 rescued samples. We observed highly motile mitochondria in nurse cells, and synchronized mitochondria movement from the nurse cells to the oocyte, highly resembling mitochondrial flow in control ring canals (*Figure 4—video 1*).

In summary, we use a minus-end gliding-only motor that cannot directly transport cargoes to distinguish the microtubule gliding function from the conventional cargo transport, and show that microtubule gliding by a motor attached to the nurse cell cortex is able to drive organelle movement and thus oocyte growth.

## C-terminus of dynein light intermediate chain is sufficient for nurse cell cortical localization

Having established that cortically anchored dynein is the key to glide microtubules and deliver cargoes to the growing oocyte, we decided to investigate how the dynein complex is anchored to the nurse cell cortex. As we observed the cortical localization of Dlic-GFP in nurse cells (*Figure 4—figure supplement 1B*), we decided to make N-terminal and C-terminal truncations of Dlic (DlicNT and DlicCT; *Figure 5A*) and examine their localizations in the germ line. The Dlic N-terminus carries a GTPase-like domain and is known to interact with dynein heavy chain via a patch of conserved aromatic residues (*Schroeder et al., 2014*). The C-terminal Dlic contains the effector-binding domain that interact with BICD, Spindly and Hook-family activating adaptors to form a stable processive dynein-dynactin complex (*Lee et al., 2018*). We tagged the DlicNT and DlicCT with an optogenetic system LOVTRAP (*Wang et al., 2016*), and in dark LOVTRAP probes bring the two Dlic truncations together (*Figure 5A–B*). We found that, while DlicNT or DlicCT truncation alone does not rescue the oocyte defects caused by *Dlic-RNAi,* the DlicNT-DlicCT complex in dark is sufficient to resume the oocyte growth in *Dlic-RNAi* (*Figure 5C*). It indicates that both Dlic truncations are functional, and both are necessary for restoring the Dlic function in oocyte growth. We next examined the localizations of DlicNT and DlicCT truncations in the germ line: DlicNT appears diffused in germline cytoplasm, whereas DlicCT shows a strong cortical localization in the nurse cells (*Figure 5D–E*; *Figure 5—figure supplement 1A*).

Since the DlicCT is known to interact with dynein activating adaptors (*Lee et al., 2018*) and BicD is the most important activating adaptor for oocyte growth (*Figure 1F*), we then tested whether BicD is required for the localization of DlicCT to the nurse cell cortex. In *BicD-RNAi* background, DlicCT still localizes to the cortex and mostly evidently at the ring canal regions (*Figure 5—figure supplement 1B*), indicating that BicD is not essential for recruiting Dlic to the nurse cell cortex.

The dynactin complex includes a short actin-like filament composed of actin-related proteins (Arp1 and Arp11) and β-actin that can potentially link dynein to the cell cortex. Previous studies in mitosis revealed that the dynactin complex facilitates dynein localization at the cell cortex for spindle pulling and positioning (*Moore et al., 2008*; *Okumura et al., 2018*). Therefore, we examined the DlicCT localization in the dominant negative p150$^{Glued}$/DCTN1 mutant (*p150$^{Glued}$ΔC*) and found that the DlicCT localization is still predominantly cortical with the inhibition of the dynactin complex (*Figure 5—figure supplement 1C*). On the other hand, Lis1 has been shown to localize at the oocyte cortex and it is required to recruit dynein to the oocyte cortex (*Swan et al., 1999*). However, we found that the cortical localization of DlicCT in nurse cells is not affected by *Lis1-RNAi* (*Figure 5—figure supplement 1D*).

Another potential linker could anchor dynein to the cell cortex is the actin-microtubule crosslinker, Short stop (Shot). Previously, we have shown that Shot is localized to the nurse cell cortex and knockdown of *shot* causes oocyte growth defects (*Lu et al., 2021*). Therefore, we examined the DlicCT localization in the strongest *shot* loss-of-function mutant (*shot-RNAi* in *shot* heterozygote background), and found that DlicCT still remains cortically associated (*Figure 5—figure supplement 1E*).

In conclusion, Dlic C-terminus could facilitate the dynein complex to localize to the nurse cell cortex, independent of the dynein activating components (BicD, dynactin and Lis1) and the actin-microtubule crosslinker Shot.

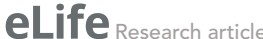

**Figure 5.** The dynein complex is tethered to the nurse cell cortex via Dlic and glides microtubules to create cytoplasmic advection to the growing oocyte. (**A**) A schematic illustration of *Drosophila* Dlic and Dlic truncations (DlicNT and DlicCT). C-terminus of DlicNT and N-terminus of DlicCT are tagged with the LOVTRAP probes, LOV2 and Zdk1, respectively. LOV2 interacts with Zdk1 in dark, and dissociates from Zdk1 in the presence of blue light. (**B**) Zdk1-DlicCT interacts with DlicNT-LOV2 in dark. GFP-binder was used to pull down Zdk1-DlicCT in cell extracts from HEK293T cells expressing both DlicNT and DlicCT constructs, and anti-Myc antibody was used to probe for Dlic truncations. Lane 1: cell extract; Lane 2: GFP-binder pulldown sample in light; lane 3; GFP-binder pulldown sample in dark. The raw unedited blots can be found in *Figure 5—source data 1*: (1) the raw pulldown blot; (2) the labeled pulldown blot. (**C**) Coexpression of DlicNT and DlicCT truncations reuses the oogenesis defects caused by *Dlic-RNAi*, while DlicNT

*Figure 5 continued on next page*

*Figure 5 continued*

truncation or DlicCT truncation alone fails to rescue. The expression is driven by one copy of *nos-Gal4-VP16*. (**D–E**) DlicNT and DlicCT localizations in the germ line. The expression is driven by one copy of *maternal αtub-Gal4[V37]*. A small Z-projection (~5 µm) is used to show the DlicNT localization, while a whole ovariole z-projection ( ~ 40 µm) is used to show the DlicCT cortical localization. To note: overexpression of DlicNT driven by *maternal αtub-Gal4[V37]* results in delayed oocyte growth. Scale bars, 50 µm. (**F**) The model of cortically anchored dynein transferring cytoplasmic contents to the growing oocyte via gliding microtubules. Dynein light intermediate chain (Dlic) recruits dynein heavy chain to the cell cortex, while the Dynactin/p150 complex, BicD and Lis1 are required for dynein activation. See also *Figure 5—video 1*.

The online version of this article includes the following video, source data, and figure supplement(s) for figure 5:

**Source data 1.** Zdk1-DlicCT interacts with DlicNT-LOV2 in dark.

**Figure supplement 1.** Cortical localization of Dlic C-terminus is independent of BicD, Dynactin/p150, Lis1 and Shot.

**Figure 5—video 1.** Cortical dynein glides microtubules and transfers cytoplasm to the *Drosophila* oocyte.
https://elifesciences.org/articles/75538/figures#fig5video1

Finally, we propose that Dlic links the dynein heavy chain to the cell cortex and thus essential for microtubule gliding in nurse cells. Gliding microtubules create cytoplasmic advection within nurse cells and through the ring canals, bringing cytoplasmic contents from nurse cells to the growing oocyte (*Figure 5F*; *Figure 5—video 1*). It remains to be determined which component(s) of the cortical network in nurse cells interact with Dlic and recruit the dynein complex to the cell cortex.

## Discussion

As the main microtubule minus-end directed motor in animal cells, cytoplasmic dynein is responsible for a wide variety of cellular functions, ranging from cell division to intracellular transport. In *Drosophila* ovary, dynein plays an essential role in nurse cell-to-oocyte transport of mRNAs and organelles (*Mische et al., 2007*; *Clark et al., 2007*; *Nicolas et al., 2009*; *Lu et al., 2021*). This was logically attributed to the conventional mode of dynein-driven transport: the motor attached to the cargo moves on microtubule tracks located inside ring canals and carries cargoes to the oocyte (*Figure 3A*).

In this study, we reveal a novel mechanism of bulk cargo transport by cytoplasmic dynein. First, we show that dynein core components and its regulatory cofactors are required for *Drosophila* oocyte growth (*Figure 1*). By imaging microtubules in live ovaries, we demonstrate that microtubules are actively moved by dynein from nurse cells to the growing oocytes (*Figure 2*). Furthermore, we use an artificial cargo that does not bind motors and show that direct dynein-cargo interaction is not necessary for the cargo movement in nurse cells or its transporting to the oocyte (*Figure 3*), supporting a 'go-with-the-flow' mechanism driving cytoplasm transfer from nurse cells to the oocyte. To distinguish cytoplasmic advection from conventional cargo transport, we build a chimeric gliding-only motor by anchoring a minus-end-directed plant kinesin, Kin14VIb, to the cortex, and find that this chimeric cortically localized motor is sufficient to drive organelle transport and oocyte growth (*Figure 4*). Lastly, we identify that the C-terminus of Dlic is sufficient to target the dynein complex to the nurse cell cortex (*Figure 5*). Therefore, we propose a novel mechanism of dynein for bulk cargo transport: cortically anchored dynein glides microtubules in the nurse cells; in turn these gliding microtubules move cytoplasmic contents within the nurse cells and from the nurse cells to the oocyte through the ring canals to the growing oocyte (*Figure 5F*; *Figure 5—video 1*).

### A novel phase of nurse cell-to-oocyte transport

Previously, nurse cell-to-oocyte transport has been divided into two phases: the early slow selective phase and the late fast non-selective phase (*Mahajan-Miklos and Cooley, 1994*; *Buszczak and Cooley, 2000*). The early phase is characterized by dynein-driven cargo transport along microtubules to the oocyte (*Mische et al., 2007*; *Clark et al., 2007*; *Nicolas et al., 2009*; *Lu et al., 2021*). The late massive nurse cell-to-oocyte transport phase is known as nurse cell dumping, which occurs at late stage 10B to stage 11 (*Mahajan-Miklos and Cooley, 1994*; *Buszczak and Cooley, 2000*).

Here, we report a new phase of cytoplasmic advection from nurse cells to the oocyte driven by microtubules that are transported by cortical dynein. This occurs between the two previously described phases, slow selective transport, and fast nurse dumping. As microtubules can drag adjacent contents in the viscous cytoplasm (*Lu et al., 2016*; *Lu et al., 2018*), we believe that this transport is non-selective. This conclusion is supported by the fact that the neutral particles (GEMs) are

moved in the nurse cells and concentrated in the oocyte by dynein. Furthermore, we have found that a chimeric gliding-only motor unrelated to dynein that cannot directly transport cargoes along microtubules, plant kinesin-14, is able to rescue mitochondria transport from the nurse cells to the oocyte (*Figure 4—video 1*). As we use the motor domain of a moss kinesin-14 that has no known homology with the motor proteins that interact the mitochondrial adaptor protein Milton (e.g. KHC and Myosin 10 A) (*Glater et al., 2006*; *Liu et al., 2008*), we conclude that the advection created by microtubule movement in the ring canals is not cargo-specific, and it is different from the early selective transport phase.

During the dumping phase, nurse cells 'squeeze' all the cytoplasmic contents to the oocyte, which is caused by non-muscle myosin-II contraction (*Gutzeit, 1986*; *Jordan and Karess, 1997*), and associated with fast cytoplasmic streaming occurring in the oocyte mixing the dumped contents with the ooplasm (*Mahajan-Miklos and Cooley, 1994*; *Lu et al., 2018*; *Gutzeit and Koppa, 1982*). We reason that the dynein-driven microtubule flow we observed is distinct from nurse cell dumping: (1) it occurs prior to nurse cell dumping and ooplasmic streaming (stages 8~9 versus stages 10B-11); (2) the nurse cell size grows drastically during microtubule flow stages (*Figure 2—figure supplement 2A*), instead of fast shrinking associated with nurse cell dumping; (3) it requires cytoplasmic dynein, instead of myosin-II activity (*Figure 2—figure supplement 2*; *Figure 2—videos 6; 9*); (4) the 'dumpless' mutants develop normally without major oocyte growth defects to stages 9~10 (*Wheatley et al., 1995*; *Jordan and Karess, 1997*; *Doerflinger et al., 2022*; *Cooley et al., 1992*; *Myster et al., 2000*; *Imran Alsous et al., 2021*), which is noticeably different from the small oocyte phenotype we observed in dynein knockdown (*Figure 1*).

The dynein-driven microtubule flow occurs concurrently with the most rapid oocyte growth at stage 9 (*Figure 2I*). During stage 9, the oocyte size can increase up to 10 X within 6 hr (*Figure 2—figure supplement 1B–1C*). To achieve such a rapid growth, each of the four nurse cell-oocyte ring canals with a diameter of ~6.5 μm is estimated to have a flow of ~150 nm/s throughout the whole time. This is consistent with our experimental measurements of the velocities of both microtubules and small spherical particles in bright-filed images (*Figure 2H*). Noticeably, mitochondria move at a slightly lower velocity compared to microtubules and small spherical particles, which is expected for organelles and large particles that are moved by viscous drag created by microtubule gliding (*Figure 2H*).

Microtubule movement in the nurse cells can be detected as early as in stage 6 egg chambers; however, the fast microtubule movement through the nurse cell-oocyte ring canals only starts in late stage 8 egg chambers, and becomes robust in stage 9 egg chambers. The stage-specific microtubule passage through the ring canals could be controlled by microtubule organization within the ring canals. In early oogenesis (stages 1–6), microtubules are nucleated from the oocyte and grow into the nurse cells with their plus-ends (*Theurkauf, 1994*; *Nashchekin et al., 2021*). Even at stages 7–8, the microtubules within the ring canals still contain more plus-ends in the nurse cells and more minus-ends in the oocyte (*Lu et al., 2021*). This particular microtubule orientation allows efficient dynein-dependent conventional cargo transport to the oocyte (*Figure 3A*), but restricts dynein-dependent microtubule movement from nurse cells to the oocyte. Hence, these microtubules in the ring canals could basically function as a plug, and as a result microtubule flow through the ring canals is not often seen at these stages (*Figure 2—video 2*). Massive microtubule reorganization occurs during mid-oogenesis (stages 7–8) and creates an anterior-posterior microtubule gradient in the oocyte with more microtubules minus-ends anchored at the anterior and lateral cortex at stages 8-9 (*Lu et al., 2020*; *Theurkauf, 1994*; *Nashchekin et al., 2016*). This microtubule reorganization in the oocyte could potentially disassemble the original 'wrong' orientated microtubules in the nurse cell-oocyte ring canals, which facilitates the dynein-driven microtubule flow to the oocyte (*Figure 2—video 2*; *Figure 3B*).

To summarize, the dynein-microtubule driven bulk cargo flow from nurse cells to the oocyte presents a novel phase between early selective transport and late non-selective dumping, which is essential for *Drosophila* oocyte growth. The stage-specific microtubule flow through the ring canals suggests that the complete small oocyte phenotype we observed in the dynein mutants are probably caused by the combination of lack of directed transport along microtubules at early stages and absence of microtubule-driven cytoplasmic advection to the oocyte at stage 9.

## Dynein anchorage at the cell cortex

In this study, we found that cortical dynein glides microtubules to create local cytoplasmic advection and move cargoes to the growing oocyte. Furthermore, we found that the C-terminus of dynein light intermediate chain (DlicCT) targets the dynein complex to the nurse cell cortex. DlicCT contains the effector-binding domain that interacts with multiple dynein activating adaptors (*Lee et al., 2018*). Recently, our lab reported that Spindly, a dynein activating adaptor that interacts with DlicCT (*Lee et al., 2018*) and recruits dynein to the kinetochore in mitosis (*Clemente et al., 2018*), anchors dynein to cortical actin in axons, thus pushing microtubules of the wrong polarity out of the axons in *Drosophila* neurons (*Del Castillo et al., 2020*). However, knockdown of Spindly does not disrupt dynein-dependent oocyte growth (*Figure 1F*). Thus, it indicates that the ovary uses a different specific mechanism for dynein cortical targeting. We showed that cortically recruited BicD is able to rescue the oocyte growth arrest caused by *BicD-RNAi* (*Figure 4A–D*), implying that BicD could contribute to the linkage between dynein and the cortex. Other studies also have suggested that the dynactin complex and Lis1 are involved in dynein cortical anchorage (*Moore et al., 2008*; *Okumura et al., 2018*; *Swan et al., 1999*). Nevertheless, DlicCT cortical localization is unaffected after inhibition of BicD, dynactin/p150, and Lis1 (*Figure 5—figure supplement 1*), indicating Dlic uses a novel mechanism of anchoring dynein to the cortex in nurse cells. Intriguingly, Dlic interacts with the *Drosophila* Par-3 homolog, Bazooka (Baz), and PIP5Kinase Skittles (SKTL) in the ovary (*Jouette et al., 2019*), while Baz and SKTL are known to localize at the nurse cell cortex (*Jouette et al., 2019*; *Doerflinger et al., 2010*). In vertebrates, dynein light intermediate chain 2 interacts with Par-3, 14-3-3 ε and ζ to anchor the dynein complex at the cortex for regulating mitotic spindle orientation (*Mahale et al., 2016*). Furthermore, NuMA and its *Drosophila* homology Mud link the dynein complex to the cortex via the interaction with membrane localized Gαi, for pulling mitotic spindle poles in dividing cells (*Kotak and Gönczy, 2013*). Recently, it has been shown that NuMA has a Hook domain and a CC1-Box-like motif, both of which interact with the effector-binding domain of DLIC (*Renna et al., 2020*). Therefore, Baz/Par-3, SKTL, 14-3-3 ε and ζ, and NuMA/Mud are all potential linkers of Dlic to the cell cortex. However, loss of SKTL and Mud, mutants of 14-3-3 ε and ζ, or displacement of Baz from the cortex does not result in apparent oocyte growth defects, distinct from the dynein mutants (*Jouette et al., 2019*; *Doerflinger et al., 2010*; *Benton et al., 2002*; *Yu et al., 2006*). It is possible that these linkers are genetically redundant with each other to ensure the correct cortical localization of the dynein complex. Thus, disruption of one of the linkers does not cause the dynein complex to fall off from the cortex. Alternatively, Dlic is localized to the cortex independently of Par-3/Baz, SKTL, 14-3-3, and NuMA/Mud, by a new mechanism that awaits further studies.

Furthermore, we consistently observe cytoplasmic advection from the nurse cells to the oocyte in stage 9 egg chambers, suggesting a direction-controlling mechanism underlying the persistent oocyte growth. We speculate that the flow directionality could be attributed to different levels of dynein gliding activity. The dynein complex is strongly localized to the nurse cell cortex, but to a much lesser extent to the oocyte cortex (*Figure 4—figure supplement 1*). Consistent with the dynein localization, microtubules are more cortically localized in the nurse cells than in the oocyte (*Figure 2E*). This differences in dynein localization and microtubule organization may result in a higher dynein-driven microtubule gliding activity in the nurse cells, and therefore creates the directional flow through the ring canals to the growing oocyte.

In addition, this directionality could be controlled by the gatekeeper protein Short stop (Shot). Recently, we demonstrated that Shot is asymmetrically localized at actin fibers of ring canals on the nurse cell side and controls the cargo transport direction between nurse cells and the oocyte (*Lu et al., 2021*). Given the nature of Shot's actin-microtubule crosslinking activity, it could serve as an organizer of microtubules along the actin filaments of the ring canals on the nurse cell side and facilitate their transport toward the oocyte.

## The 'go-with-the-flow' mechanism of cytoplasmic transport

Here we report that the minus-end directed motor, cytoplasmic dynein, glides microtubules, and microtubules in turn drag the surrounding highly viscous cytoplasm and transfer cytoplasmic contents from nurse cells to the oocyte. To our knowledge, this is the first report of microtubule gliding by cortical dynein driving cargo movement in interphase cells (the 'go-with-the-flow' mechanism).

Previously, we have demonstrated a similar mechanism for another major microtubule motor, conventional kinesin (kinesin-1). We have shown that kinesin-1 can slide microtubules against each other (*Lu et al., 2013b*; *Lu et al., 2015*; *Jolly et al., 2010*; *Barlan et al., 2013*; *Del Castillo et al., 2015*), and this microtubule sliding drives ooplasmic streaming, bulk circulation of the entire cytoplasm in late-stage oocytes that is essential for localization of the posterior determinant, *osk*/Staufen RNPs (*Lu et al., 2016*; *Lu et al., 2018*). Thus, both major microtubule motors, plus-end directed kinesin-1 and minus-end directed dynein, in addition to the canonical mode of cargo transport along microtubules, can drive bulk transport of viscous cytoplasm. Yet kinesin-1 and dynein drive bulk movement in different manners: kinesin-1 drives microtubule sliding against each other, while dynein glides microtubules along the cortex; and for different purposes: kinesin-1 powers intracellular circulation, whereas dynein propels intercellular transport.

This 'go-with-the-flow' mechanism is highly efficient for bulk cargo delivery, especially in large cells (*Goldstein and van de Meent, 2015*), such as the oocyte. The dynein-driven cytoplasmic advection allows the oocyte to acquire cytoplasmic materials for its rapid growth. Interestingly, the ring canals have been observed in female germline cells of vertebrate organisms (e.g. human, rabbit, rat, hamster, mouse, chicken, and frog) (*Haglund et al., 2011*). Particularly, it has been shown that cytoplasmic contents such as mitochondria and Golgi material are transferred to the mouse oocyte from interconnected cyst cells in a microtubule-dependent fashion (*Lei and Spradling, 2016*; *Niu and Spradling, 2021*) . As dynein is highly conserved across species, it is important to have further studies to examine whether it plays a similar role in germline cytoplasmic transfer in higher organisms.

## Materials and methods
### *Drosophila* strains

Fly stocks and crosses were maintained on standard cornmeal food (Nutri-Fly Bloomington Formulation, Genesee, Cat #: 66–121) supplemented with dry active yeast at room temperature (~24–25°C). The following fly stocks were used in this study: *mat αtub-Gal4*[V37] (III, Bloomington *Drosophila* Stock Center #7063); *Act5C-Gal4* (III, Bloomington *Drosophila* Stock Center #3954); *nos-Gal4-VP16* (III, from Dr. Edwin Ferguson, the University of Chicago *Van Doren et al., 1998*; *Lu et al., 2012*); *UAS-Dhc64C-RNAi* (line #1: TRiP.GL00543, attP40, II, Bloomington *Drosophila* Stock Center #36583, targeting *DHC64C* CDS 10044–10,064 nt, 5'-TCGAGAGAAGATGAAGTCCAA-3'; line #2: TRiP.HMS01587, attP2, III, Bloomington *Drosophila* Stock Center #36698, targeting *DHC64C* CDS 1302–1,322 nt, 5'-CCGA GACATTGTGAAGAAGAA-3') (*Lu et al., 2015*; *del Castillo et al., 2015*); *UASp-Gl*[ΔC] (1–826 residues of p150[Glued]/*DCTN1*, based on the *Glued*[1] domintant mutation) (II, 16.1, from Dr. Thomas Hays, University of Minnesota) (*Mische et al., 2007*); *UAS-Lis1-RNAi* (II, from Dr. Graydon Gonsalvez, Augusta University, targeting *Lis1* CDS 1197–1217 nt, 5'-TAGCGTAGATCAAACAGTAAA-3') (*Liu et al., 2015*); *UAS-BicD-RNAi* (TRiP.GL00325, attP2, III, Bloomington *Drosophila* Stock Center #35405, targeting *BicD* 3'UTR 639–659 nt, 5'-ACGATTCAGATAGATGATGAAA-3'); *UAS-Spindly-RNAi* (TRiP.HMS01283, attP2, III, Bloomington *Drosophila* Stock Center #34933, targeting *Spindly* CDS 1615–1,635 nt, 5'-CAGGACGCGGTTGATATCAAA-3') (*Del Castillo et al., 2020*); *UAS-hook-RNAi* (TRiP.HMC05698, attP40, II, Bloomington *Drosophila* Stock Center #64663, targeting *HOOK* CDS 241–261 nt, 5'-TACGACTACTACAGCGACGTA-3'); *UAS-GFP-RNAi* (attP2, III, Bloomington *Drosophila* Stock Center #41551); *UASp-tdMaple3-αtub84B* (II) (*Lu et al., 2016*); *UASp-EMTB-3XTagRFP* (III) (*Lu et al., 2020*); *mat αtub67C-EMTB-3XGFP-sqh 3'UTR* (attP40, an unpublished gift from Yu-Chiun Wang lab, RIKEN Center for Biosystems Dynamics Research); *UASp-GFP-Patronin* (II) (from Dr. Uri Abdu, Ben-Gurion University of the Negev) (*Lu et al., 2020*; *Lu et al., 2018*; *Baskar et al., 2019*); *Jupiter-GFP* (protein trap line ZCL2183, III) (*Lu et al., 2013b*; *Lu et al., 2015*; *Karpova et al., 2006*); *UASp-LifeAct-TagRFP* (III, 68E, Bloomington *Drosophila* Stock Center # 58714); *UASp-F-Tractin-tdTomato* (II, Bloomington *Drosophila* stock center #58989) (*Spracklen et al., 2014*); *UAS-zip-RNAi* (TRiP.GL00623, attP40, II, Bloomington *Drosophila* Stock Center #37480, targeting *Zip* 3'UTR 36–56 nt, 5'-CAGGAAGAAGGT GATGATGAA-3'); *hs-Flp*[12] (X, Bloomington *Drosophila* Stock Center #1929); *FRTG13 ubi-GFP.nls* (II, Bloomington *Drosophila* Stock Center # 5826); *FRTG13 zip*[2] /*CyO* (II, Bloomington *Drosophila* stock center # 8739); *Sqh-GFP* (III, Bloomington *Drosophila* stock center #57145); *UASp-Mito-MoxMaple3* (II, and III) (*Lu et al., 2021*); *ubi-GFP-Pav* (II, from Dr. David Glover, Caltech) (*Minestrini et al., 2002*); *UASp-RFP-Golgi* (II, Bloomington *Drosophila* Stock Center # 30908, aka *UASp-GalT-RFP*) (*Chowdhary*

*et al., 2017*); *pDlic-Dlic-GFP* (II, under the control of its native promoter, from Dr. Thomas Hays, University of Minnesota) (*Neisch et al., 2021*); *shot*[ΔEGC] (from Dr. Ferenc Jankovics, Institute of Genetics, Biological Research Centre of the Hungarian Academy of Sciences) (*Takács et al., 2017*); *UAS-shot*[EGC]*-RNAi* (in pWalium22 vector, inserted at attP2, III) (*Lu et al., 2021*). The following fly stocks were generated in this study using either PhiC31-mediated integration or P-element-mediated transformation: *UASp-Dlic-RNAi* (targeting *Dlic* 3'UTR 401–421 nt, 5'-AGAAATTTAACAAAAAAAAAA –3', in pWalium22 vector, inserted at attP-9A (VK00005) 75A10 site, III, M5); *UASp-GEM* (III, M1); *UASp-F-Tractin-Myc-BicD* (II, M4); *UASp-F-Tractin-GFP-Kin14VIb* (II, M1); *UASp-Myc-HA-DlicNT-LOV2* at attP14 (36A10, II, M2); *UASp-Zdk1-DlicCT-sfGFP-Myc* at attP33 (50B6, II, M2).

## Plasmid constructs

### pWalium22-Dlic3'UTR-shRNA

The oligos of Dlic3'UTR-shRNA (agt<u>AGAAATTTAACAAAAAAAAAA</u>tagttatattcaagcata<u>TTTTTTTT</u> <u>TTGTTAAATTTCT</u>gc) were synthesized and inserted into the pWalium22 vector (*Drosophila* Genomics Resource Center, Stock Number #1473, 10XUAS) (*Ni et al., 2011*) by NheI(5')/EcoRI(3').

### pUASp-GEM

GEM (PfV-GS-Sapphire) was cut from pCDNA3.1-pCMV-PfV-GS-Sapphire (Addgene plasmid # 116933; RRID:Addgene_116933) (*Delarue et al., 2018*), and inserted into the pUASp vector by NheI(5')/XbaI(3').

### pUASp-F-Tractin-Myc-BicD

F-Tractin (the actin binding domain of rat Inositol trisphosphate 3-kinase A (lTPKA), residues 9–40, atgGGCATGGCGCGACCACGGGGCGCGGGGCCCTGCAGCCCCGGGTTGGAGCGGGCTCCGCGC CGGAGCGTCGGGGAGCTGCGCCTGCTCTTCGAA) (*Spracklen et al., 2014*; *Johnson and Schell, 2009*) was synthesized and inserted into pUASp by KpnI (5')/SpeI (3'). Myc-BicD was amplified from pAC-FRB-GFP-BicD (*del Castillo et al., 2015*) and inserted into pUASp vector by SpeI (5')/XbaI (3').

### pUASp-F-Tractin-GFP-Kin14VIb

F-Tractin (the actin binding domain of rat Inositol trisphosphate 3-kinase A (lTPKA), residues 9–40, atgGGCATGGCGCGACCACGGGGCGCGGGGCCCTGCAGCCCCGGGTTGGAGCGGGCTCCGCGC CGGAGCGTCGGGGAGCTGCGCCTGCTCTTCGAA) (*Spracklen et al., 2014*; *Johnson and Schell, 2009*) was synthesized and inserted into pUASp by KpnI (5')/SpeI (3'). EGFP and Kin14VIb were amplified from the vector of pET15b-EGFP-GCN4-kinesin14VIb (a gift from Dr. Gohta Goshima, Nagoya University, Japan) (*Jonsson et al., 2015*) by PCR and inserted into pUASp by SpeI(5')/NotI(3') and NotI(5')/XbaI (3'), respectively.

### pUASp-Myc-HA-DlicNT-LOV2-attB

DlicNT (1–370 residues) and LOV2[WT] were amplified by PCR from pAC-Dlic-EGFP (a gift from Dr. Thomas Hays, University of Minnesota) (*Mische et al., 2008*) and pTriEx-NTOM20-mVenus-LOV2[WT] (a gift from Dr. Klaus Hahn, University of North Carolina at Chapel Hill) (*Wang et al., 2016*) and inserted into pMT.A vector by EcoRV(5')/XhoI(3') and XhoI(5')/XbaI(3'), respectively. Myc-HA with linkers (GGSG) was synthesized and inserted into the pMT-A-DlicNT-LOV2[WT] by EcoRI(5')/ EcoRV(3'). Myc-HA-DlicNT-LOV2[WT] was subcloned from the pMT.A-Myc-HA-DlicNT-LOV2[WT] into the pUASp vector by KpnI(5')/XbaI(3'), and attB site was subcloned from the pUASp-attB vector (*Drosophila* Genomics Resource Center/DGRC vector #1358) by AatII(5')/AatII(3') to create pUASp-Myc-HA-DlicNT-LOV2-attB.

### pUASp-Zdk1-DlicCT-sfGFP-Myc-attB

Zdk1 and C-terminus of Dlic (371–493 residues) were amplified by PCR from pTriEx-mCherry-Zdk1 (a gift from Dr. Klaus Hahn, University of North Carolina at Chapel Hill) (*Wang et al., 2016*) and pAC-Dlic-EGFP (a gift from Dr. Thomas Hays, University of Minnesota) (*Mische et al., 2008*), and inserted into pMT.A by SpeI(5')/EcoRI(3') and EcoRI(5')/EcoRV(3'), respectively. Superfolder GFP (sfGFP) with a Myc tag was inserted into the pMT.A-Zdk1-DlicCT by EcoRV(5')/XbaI(3'). Zdk1-DlicCT-sfGFP-myc

was then subcloned into the pUASp-attB vector (*Drosophila* Genomics Resource Center/DGRC vector #1358) by SpeI(5')/XbaI(3') to create pUASp-Zdk1-DlicCT-sfGFP-Myc-attB.

### pcDNA3.1(+)-Myc-HA-DlicNT-LOV2

Myc-HA-DlicNT-LOV2 was subcloned from pMT.A-Myc-HA-DlicNT-LOV2[WT] into pcDNA3.1(+) by KpnI(5')/XbaI(3').

### pcDNA3.1(+)-Zdk1-DLicCT-sfGFP-Myc

Zdk1-DlicCT-sfGFP-Myc was amplified by PCR from pMT.A-Zdk1-DlicCT-sfGFP-Myc and inserted into pcDNA3.1(+) by HindIII(5')/XbaI(3').

## Immunostaining of *Drosophila* egg chambers

A standard fixation and staining protocol was previously described (*Lu et al., 2021*; *Lu et al., 2016*; *Lu et al., 2020*; *Lu et al., 2018*; *Lu et al., 2012*). Samples were stained with primary antibody at 4 °C overnight and with fluorophore-conjugated secondary antibody at room temperature (24~25 °C) for 4 hr. Primary antibody used in this study: mouse monoclonal anti-Orb antibody (Orb 4H8, Developmental Studies Hybridoma Bank, supernatant, 1:5); rabbit phospho-MLC (pMLC 2/Ser19, 1:100, Cell Signaling, Cat# 3671); mouse monoclonal anti-DHC antibody (2C11-2, Developmental Studies Hybridoma Bank, concentrate, 1:50); mouse monoclonal anti-BicD antibody (anti-Bicaudal-D 4C2, Developmental Studies Hybridoma Bank, supernatant, 1:5); mouse monoclonal anti-Myc antibody (1-9E10.2, 1:100) (*Karcher et al., 2001*). Secondary antibody used in this study: FITC-conjugated or TRITC-conjugated anti-mouse secondary antibody (Jackson ImmunoResearch Laboratories, Inc; Cat# 115-095-062 and Cat# 115-025-003) at 10 µg/ml; FITC-conjugated anti-rabbit secondary antibody (Jackson ImmunoResearch Laboratories, Inc; Cat#111-095-003) at 10 µg/ml.

Some samples were stained with rhodamine-conjugated phalloidin (0.2 µg/ml) and DAPI (1 µg/mL) for 1 ~ 2 hr before mounting. Samples were imaged on a Nikon A1plus scanning confocal microscope with a GaAsP detector and a 20 × 0.75 N.A. lens using Galvano scanning, a Nikon W1 spinning disk confocal microscope (Yokogawa CSU with pinhole size 50 µm) with a Photometrics Prime 95B sCMOS Camera or a Hamamatsu ORCA-Fusion Digital CMOS Camera and a 40 × 1.30 N.A. oil lens or a 40 × 1.25 N.A. silicone oil lens, or a Nikon Eclipse U2000 inverted stand with a Yokogawa CSU10 spinning disk confocal head, a Photometrics Evolve EMCCD camera and a 40 × 1.30 N.A. oil lens, all controlled by Nikon Elements software. Z-stack images were acquired every 1 µm/step for whole ovariole imaging or 0.3~0.5 µm/step for individual egg chambers.

## Microtubule staining in *Drosophila* egg chambers

Ovaries were dissected in 1 X Brinkley Renaturing Buffer 80 (BRB80, 80 mM piperazine-N,N'-bis(2-ethanesulfonic acid) [PIPES], 1 mM MgCl2, 1 mM EGTA, pH 6.8) and fixed in 8% EM-grade formaldehyde +1 X BRB80 +0.1% Triton X-100 for 20 min on the rotator; briefly washed with 1 X PBTB (1 X PBS + 0.1% Triton X-100 +0.2% BSA) five times and stained with FITC-conjugated β-tubulin antibody (ProteinTech, Cat# CL488-66240) 1:100 at 4 °C overnight; then samples were stained rhodamine-conjugated phalloidin and DAPI for 1 hr before mounting. Samples were imaged using a Nikon W1 spinning disk confocal microscope (Yokogawa CSU with pinhole size 50 µm) with a Photometrics Prime 95B sCMOS Camera, and a 40 × 1.25 N.A. silicone oil lens, controlled by Nikon Elements software. Images were acquired every 0.3 µm/step in z stacks and 3D deconvolved using Richardson-Lucy iterative algorithm provided by Nikon Elements.

## Live imaging of *Drosophila* egg chamber

Young mated female adults were fed with dry active yeast for 16~18 hr and then dissected in Halocarbon oil 700 (Sigma-Aldrich, Cat# H8898) as previously described (*Lu et al., 2021*; *Lu et al., 2016*; *Lu et al., 2020*; *Lu et al., 2018*). Fluorescent samples were imaged using a Nikon W1 spinning disk confocal microscope (Yokogawa CSU with pinhole size 50 µm) with a Photometrics Prime 95B sCMOS Camera or a Hamamatsu ORCA-Fusion Digital CMOS Camera, and a 40 × 1.30 N.A. oil lens or a 40 × 1.25 N.A. silicone oil lens, controlled by Nikon Elements software.

## Photoconversion of tdMaple3-tubulin and Mito-MoxMaple3 in ovary

Photoconversions of tdMaple3-tubulin and Mito-MoxMaple3 were performed using illumination from a Heliophor 89 North light in the epifluorescence pathway by a 405 nm filter, either locally (for tdMaple3-tubulin) or globally (for Mito-MoxMaple3) through an adjustable pinhole in the field diaphragm position for 10~20 s. Samples were imaged either on a Nikon Eclipse U2000 inverted stand with a Yokogawa CSU10 spinning disk confocal head with a Photometrics Evolve EMCCD camera and a 40 × 1.30 N.A. oil lens, or a Nikon W1 spinning disk confocal microscope (Yokogawa CSU with pinhole size 50 µm) with a Hamamatsu ORCA-Fusion Digital CMOS Camera and a 40 × 1.25 N.A. silicone oil lens, all controlled by Nikon Elements software.

## Induction of *zip2* germline clones

*FRTG13 zip²/CyO* virgin female flies were crossed with males carrying *hs-Flp¹²/y; FRTG13 ubi-GFP.nls/CyO*. From the cross, young pupae at day 7 and day 8 AEL (after egg laying) were subjected to heat shock at 37 °C for 2 hr each day. Non CyO F1 females were collected 3–4 days after heat shock and fattened with dry active yeast overnight before dissection for fixation and Orb staining.

## Extraction and fixation of ovary samples

Ovaries were dissected and gently teased apart in 1 X BRB80. The dissected samples were extracted in 1 X BRB80 +1% Triton X-100 for 20 min without agitation. After the extraction, the samples were fixed with 8% EM-grade formaldehyde in 1 X BRB80 +0.1% Triton X-100 for 20 min on rotator, washed with 1 X PBTB (1 X PBS + 0.1% Triton X-100 +0.2% BSA) five times before immunostaining.

## Measurement of velocity in the nurse cell-oocyte ring canals

Velocities of microtubules, small particles and mitochondria were measured based on kymographs generated along lines within the nurse cell-oocyte ring canals (10~15 µm line length, and ~5 µm line width) using the MultipleKymograph plugin in FIJI (*Schindelin et al., 2012*). Velocities were calculated on these kymographs using Kymograph (time space plot) Plugin for ImageJ, written by J. Rietdorf (FMI Basel) and A. Seitz (EMBL Heidelberg) (https://www.embl.de/eamnet/html/body_kymograph.html).

## Measurement of nurse cell and oocyte size

Z stacks of triple color images of the ovarioles from *yw; ubi-GFP-Pav* stained with rhodamine conjugated-phalloidin and DAPI were acquired, and nurse cell area and oocyte area were specified (at the largest cross-section) and measured by manual polygon selection (area size) in FIJI (*Schindelin et al., 2012*).

## S2R+ cell transfection and imaging acquisition

*Drosophila* S2R + cells (DGRC Stock Number: 150) were transfected with 0.5 µg DNA (pAC-Gal4 + pUASp GEM, or pAC-LAMP1-GFP, *Minin et al., 2006*) in 12-well plate using Effectene transfection kit (Qiagen, Cat. # / ID: 301425) and then were plated 48 hr after transfection on concanavalin A-coated glass coverslips (*Lu et al., 2013a*). Cells were imaged using a Nikon W1 spinning disk confocal microscope (Yokogawa CSU with pinhole size 50 µm) with a Photometrics Prime 95B sCMOS Camera, and a 100 × 1.45 N.A. oil lens, controlled by Nikon Elements software.

## Measurement of GEM particle movement in control and *Dynein-RNAi*

Samples expressing GEM particles (*yw; ; mat atub-Gal4[V37]/UASp-GEM* and *yw; UAS-Dhc64C-RNAi/+; mat atub-Gal4[V37]/UASp-GEM*) were imaged on a Nikon W1 spinning disk confocal microscope (Yokogawa CSU with pinhole size 50 µm) with a Photometrics Prime 95B sCMOS Camera and a 40 × 1.25 N.A. silicone oil lens for 2 min at the frame rate of every 2 s, controlled by Nikon Elements software. Images were processed in Fiji and analyzed DiaTrack 3.04 Pro (*Vallotton and Olivier, 2013*), with a maximum particle jump distance of 3.2 µm/s.

## HEK293T cell transfection and GFP-binder pulldown assay

HEK293T cells (American Type Culture Collection/ATCC, CRL-3216) were co-transfected with pcDNA3.1(+)-myc-HA-DlicNT-LOV2 and pcDNA3.1(+)-Zdk1-DlicCT-sfGFP-Myc by Calcium Phosphate

transfection as previously described (*Kingston et al., 2003*). 40~48 hr later, cell extracts from transfected cells were made with red-only light on (no blue light, considered as 'dark'): cells were rinsed twice with 1 X PBS and scraped into 350 μl extraction buffer (10 mM Tris-buffer, pH 7.4; 50 mM NaCl; 0.5 mM MgCl2); cracked in cell cracker and collected in 1.5 ml microtube. Triton X-100 was added to the extract to final concentration of 1%. The extract was centrifuged at 14 k rpm for 15 min at 4 °C (Eppendorf, 5804 R). Supernatant was collected and divided equally in two parts. Both samples were mixed with 25 μl pre-washed GFP-binder agarose beads (Chromotek) (*Robert et al., 2019*). One sample was kept in aluminum foil for 2 hr at 4°C (dark); another sample was illuminated first with blue light for 1 min and then full natural light for 2 hours at 4 °C. Beads were washed twice and eluted with 40–50 μl 5 x Sample Buffer. Initial cell extract and both eluted samples were separated by electrophoresis in 8% SDS-PAGE gel, and transferred to nitrocellulose membrane (Odyssey Nitrocellulose Membrane, LI-COR). Membrane was blotted with mouse monoclonal anti-Myc antibody (1-9E10.2, 1:1000), followed by HRP-conjugated goat-anti-mouse secondary antibody (Jackson ImmunoResearch Laboratories, Cat# 115-035-003, 1:10,000). The blot was developed on Odyssey phosphorimager (LI-COR).

## Statistical analysis

The plots in figures show either percentage of phenotypes, or average values, as indicated in figure legends. Error bars represent 95% confidence intervals. N stands for number of samples examined in each assay, unless it is specified elsewhere in figure legends. Unpaired t tests with Welch's correction were performed in GraphPad Prism 8.0.2. p values and levels of significance are listed in figure legends.

## Acknowledgements

We thank many colleagues who generously contributed reagents for this work: Dr. Thomas Hays (University of Minnesota); Dr. Graydon Gonsalvez (Augusta University); Dr. Uri Abdu (Ben-Gurion University of the Negev, Israel); Dr. David Glover (Caltech); Dr. Gohta Goshima (Nagoya University, Japan); Dr. Klaus Hahn (University of North Carolina Chapel Hill); Dr. Andrew Carter (MRC Laboratory of Molecular Biology, Cambridge); Dr. Edwin Ferguson (the University of Chicago); Dr. Yu-Chiun Wang (RIKEN Center for Biosystems Dynamics Research, Japan); Bloomington *Drosophila* Stock Center (supported by National Institutes of Health grant P40OD018537) and *Drosophila* Genomics Resource Center (supported by NIH grant 2P40OD010949). The anti-Orb 4H8 monoclonal antibody developed by Dr. Paul D Schedl's group at Princeton University, anti-DHC 2C11-2 antibody developed by Dr. Jonathan M Scholey's group at University of California, Davis, and anti-BicD 4C2 antibody developed by Dr. Ruth Steward group at Waksman Institute Rutgers University, were obtained from the Developmental Studies Hybridoma Bank, created by the NICHD of the NIH and maintained at The University of Iowa, Department of Biology, Iowa City, IA 52242. We also thank all the Gelfand laboratory members for support, discussions, and suggestions. Research reported in this study was supported by the National Institute of General Medical Sciences grant R35GM131752 to VI Gelfand.

## Additional information

### Funding

| Funder | Grant reference number | Author |
| --- | --- | --- |
| National Institute of General Medical Sciences | R35 GM131752 | Vladimir I Gelfand |

The funders had no role in study design, data collection and interpretation, or the decision to submit the work for publication.

### Author contributions

Wen Lu, Conceptualization, Data curation, Formal analysis, Investigation, Methodology, Resources, Writing – original draft; Margot Lakonishok, Anna S Serpinskaya, Data curation, Investigation; Vladimir

I Gelfand, Conceptualization, Data curation, Formal analysis, Funding acquisition, Investigation, Methodology, Project administration, Resources, Supervision, Writing – original draft, Writing – review and editing

**Author ORCIDs**
Wen Lu http://orcid.org/0000-0002-8849-8100
Vladimir I Gelfand http://orcid.org/0000-0002-6361-2798

**Decision letter and Author response**
Decision letter https://doi.org/10.7554/eLife.75538.sa1
Author response https://doi.org/10.7554/eLife.75538.sa2

## Additional files

**Supplementary files**
• Transparent reporting form

**Data availability**
All data generated or analyzed during this study are included in the manuscript and supporting file; Source Data files have been provided for Figure 5.

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

# Appendix 1

## Appendix 1—key resources table

| Reagent type (species) or resource | Designation | Source or reference | Identifiers | Additional information |
|---|---|---|---|---|
| gene (*Drosophila melanogaster*) | BicD | doi: 10.7554/ eLife.10140 (***Del Castillo et al., 2015***) | CG6605 | |
| gene (*Drosophila melanogaster*) | Dlic | Thomas Hays lab (University of Minnesota); doi:10.1091/mbc. E08-05-0483 (***Mische et al., 2008***) | CG1938 | |
| gene (*Rattus norvegicus*) | Inositol trisphosphate 3-kinase A (ITPKA) | doi:10.1091/mbc. E09-01-0083 (***Johnson and Schell, 2009***); doi:10.1016 /j. ydbio.2014.06.022 (***Spracklen et al., 2014***) | Gene ID: 81677 | |
| gene (*Pyrococcus furiosus*) | PfV | Addgene plasmid # 116933; doi:10.1016 /j.cell. 2018.05.042 (***Delarue et al., 2018***) | GenBank: AB214633.1; RRID:Addgene_116933 | |
| gene (*Physcomitrella patens*) | Kin14VIb | Gohta Goshima lab; doi:10.1038/NPLANTS.2015.87 (***Jonsson et al., 2015***) | | |
| genetic reagent (*Drosophila melanogaster*) | mat αtub-Gal4[V37] | Bloomington *Drosophila* Stock Center | BDSC: 7063; FlyBase ID: FBti0016914 | |
| genetic reagent (*Drosophila melanogaster*) | Act5C-Gal4 | Bloomington *Drosophila* Stock Center | BDSC: 3954; FlyBase ID: FBti0012292 | |
| genetic reagent (*Drosophila melanogaster*) | nos-Gal4-VP16 | Edwin Ferguson lab (University of Chicago); doi: 10.1016 / s0960-9822(98)70091-0 (***Van Doren et al., 1998***) | | |
| genetic reagent (*Drosophila melanogaster*) | UAS-Dhc64C-RNAi | Bloomington *Drosophila* Stock Center | BDSC: 36583; FlyBase ID: FBti0146459; TRiP.GL00543 | |
| genetic reagent (*Drosophila melanogaster*) | UAS-Dhc64C-RNAi | Bloomington *Drosophila* Stock Center | BDSC: 36698; FlyBase ID: FBti0146710; TRiP.HMS01587 | |
| genetic reagent (*Drosophila melanogaster*) | UASp-GlΔC | Thomas Hays lab (University of Minnesota); doi:10.1091/mbc. E06-10-0959 (***Mische et al., 2007***) | Line # 16.1 | |
| genetic reagent (*Drosophila melanogaster*) | UAS-Lis1-RNAi | Graydon Gonsalvez lab (Augusta University); doi:10.1534/genetics. 115.180018 (***Liu et al., 2015***) | | |
| genetic reagent (*Drosophila melanogaster*) | UAS-BicD-RNAi | Bloomington *Drosophila* Stock Center | BDSC: 35405; FlyBase ID: FBti0144407; TRiP.GL00325 | |
| genetic reagent (*Drosophila melanogaster*) | UAS-Spindly-RNAi | Bloomington *Drosophila* Stock Center | BDSC: 34933; FlyBase ID: FBti0144908; TRiP.HMS01283 | |
| genetic reagent (*Drosophila melanogaster*) | UAS-hook-RNAi | Bloomington *Drosophila* Stock Center | BDSC: 64663; FlyBase ID: FBti0183911; TRiP.HMC05698 | |
| genetic reagent (*Drosophila melanogaster*) | UAS-GFP-RNAi | Bloomington *Drosophila* Stock Center | BDSC: 41551; FlyBase ID: FBti0148872 | |

*Appendix 1 Continued on next page*

*Appendix 1 Continued*

| Reagent type (species) or resource | Designation | Source or reference | Identifiers | Additional information |
|---|---|---|---|---|
| genetic reagent (*Drosophila melanogaster*) | *UASp-tdMaple3-αtub84B* | Our lab; doi:10.1073/pnas.1522424113 (***Lu et al., 2016***) | | |
| genetic reagent (*Drosophila melanogaster*) | *UASp-EMTB-3XTagRFP* | Our lab; doi:10.7554/eLife.54216 (***Lu et al., 2020***) | | |
| genetic reagent (*Drosophila melanogaster*) | *mat αtub67C-EMTB-3XGFP-sqh 3'UTR* | Yu-Chiun Wang lab (RIKEN Center for Biosystems Dynamics Research) (an unpublished gift) | | |
| genetic reagent (*Drosophila melanogaster*) | *UASp-GFP-Patronin* | Uri Abdu lab (Ben Gurion University); doi:10.7554/eLife.54216 (***Lu et al., 2020***); doi:10.1083/jcb.201709174 (***Lu et al., 2018***); doi:10.1242/bio.039552 (***Baskar et al., 2019***); doi:10.1016 /j.cub.2021.05.010 (***Lu et al., 2021***) | | |
| genetic reagent (*Drosophila melanogaster*) | *Jupiter-GFP* | Yale GFP Protein Trap Database doi: 10.1073/pnas.261408198 (***Morin et al., 2001***); doi:10.1016 /j.cub.2013.04.050 (***Lu et al., 2013b***); doi:10.1091/mbc.E14-10-1423 (***Lu et al., 2015***); doi:10.1002 /cm.20124 (***Karpova et al., 2006***) | ZCL2183; FlyBase ID: FBal0286259 | |
| genetic reagent (*Drosophila melanogaster*) | *UASp-LifeAct-TagRFP* | Bloomington *Drosophila* Stock Center | BDSC: 58714; FlyBase ID: FBti0164964 | |
| genetic reagent (*Drosophila melanogaster*) | *UASp-F-Tractin-tdTomato* | Bloomington *Drosophila* stock center; doi:10.1016 /j.ydbio.2014.06.022 (***Spracklen et al., 2014***) | BDSC: 58989; FlyBase ID: FBti0164816 | |
| genetic reagent (*Drosophila melanogaster*) | *UAS-zip-RNAi* | Bloomington *Drosophila* Stock Center | BDSC: 37480; FlyBase ID: FBti0146538; TRiP.GL00623 | |
| genetic reagent (*Drosophila melanogaster*) | *hs-Flp[12]; Sco/CyO* | Bloomington *Drosophila* stock center | BDSC: 1929; FlyBase ID: FBti0000784 | |
| genetic reagent (*Drosophila melanogaster*) | *FRTG13 zip[2]/CyO* | Bloomington *Drosophila* stock center | BDSC: 8739; FlyBase ID: FBti0001247 and FBal0018863 | |
| genetic reagent (*Drosophila melanogaster*) | *FRTG13 Ubi-GFP.nls* | Bloomington *Drosophila* stock center | BDSC: 5826; FlyBase ID: FBti0001247, FBti0016099 and FBti0016100 | |
| genetic reagent (*Drosophila melanogaster*) | *Sqh-GFP* | Bloomington *Drosophila* stock center | BDSC: 57145; FlyBase ID: FBti0150058 | |
| genetic reagent (*Drosophila melanogaster*) | *UASp-Mito-MoxMaple3* | Our lab; doi:10.1016 /j.cub.2021.05.010 (***Lu et al., 2021***) | | |
| genetic reagent (*Drosophila melanogaster*) | *ubi-GFP-Pav* | David Glover lab (Caltech); doi: 10.1242/jcs.115.4.725 (***Minestrini et al., 2002***) | | |

*Appendix 1 Continued on next page*

*Appendix 1 Continued*

| Reagent type (species) or resource | Designation | Source or reference | Identifiers | Additional information |
|---|---|---|---|---|
| genetic reagent (*Drosophila melanogaster*) | UASp-RFP-Golgi | BIlomington stock center; doi:10.1038 /s41598-017-05679-1 (*Chowdhary et al., 2017*); doi:10.1016 /j.cub. 2021.05.010 (*Lu et al., 2021*) | BDSC: 30908; FlyBase ID: FBti0129915 | |
| genetic reagent (*Drosophila melanogaster*) | pDlic-Dlic-GFP | Thomas Hays (University of Minnesota) (*Neisch et al., 2021*) doi: https://doi.org/ 10.1101/ 2021.09.27.462070 | | |
| genetic reagent (*Drosophila melanogaster*) | shotΔEGC | Ferenc Jankovics, Institute of Genetics, Biological Research Centre of the Hungarian Academy of Sciences; doi:10.1242/jcs.193003 (*Takács et al., 2017*) | | |
| genetic reagent (*Drosophila melanogaster*) | UASp-shot-RNAi | Our lab; doi:10.1016 / j.cub.2021.05.010 (*Lu et al., 2021*) | | |
| genetic reagent (*Drosophila melanogaster*) | UASp-Dlic-RNAi | Generated in this study (available upon request, contact vgelfand@northwestern.edu) | | |
| genetic reagent (*Drosophila melanogaster*) | UASp-GEM | Generated in this study (available upon request, contact vgelfand@northwestern.edu) | | |
| genetic reagent (*Drosophila melanogaster*) | UASp-F-Tractin-Myc-BicD | Generated in this study (available upon request, contact vgelfand@northwestern.edu) | | |
| genetic reagent (*Drosophila melanogaster*) | UASp-F-Tractin-GFP-Kin14VIb | Generated in this study (available upon request, contact vgelfand@northwestern.edu) | | |
| genetic reagent (*Drosophila melanogaster*) | UASp-Myc-HA-DlicNT-LOV2 | Generated in this study (available upon request, contact vgelfand@northwestern.edu) | | |
| genetic reagent (*Drosophila melanogaster*) | UASp-Zdk1-DlicCT-sfGFP-Myc | Generated in this study (available upon request, contact vgelfand@northwestern.edu) | | |
| cell line (*Drosophila melanogaster*) | S2R + cells | *Drosophila* Genomics Resource Center (DGRC) | Stock Number: 150; RRID:CVCL_Z831 | Tested negative for mycoplasma; authenticated by the vendor |
| cell line (*Homo sapiens*) | HEK293T cells | ATCC; doi: 10.1096/ fj.201800604R (*Robert et al., 2019*) | CRL-3216 | Tested negative for mycoplasma; authenticated by the vendor |
| transfected construct (*Drosophila melanogaster*) | pcDNA3.1(+)-Myc-HA-DlicNT-LOV2 | Generated in this study (available upon request, contact vgelfand@northwestern.edu) | Contains *Drosophila* Dlic residues 1–370 and LOVTRAP probe LOV2[WT] | For transfection of HEK293T cells |
| transfected construct (*Drosophila melanogaster*) | pcDNA3.1(+)-Zdk1-DlicCT-sfGFP-Myc | Generated in this study (available upon request, contact vgelfand@northwestern.edu) | Contains LOVTRAP probe Zdk1 and *Drosophila* Dlic residues 371–493 | For transfection of HEK293T cells |
| antibody | anti-Orb antibody (Mouse monoclonal) | Developmental Studies Hybridoma Bank (DSHB) | orb 4H8; RRID: AB_528418 | (1:5) dilution |

*Appendix 1 Continued on next page*

Appendix 1 Continued

| Reagent type (species) or resource | Designation | Source or reference | Identifiers | Additional information |
|---|---|---|---|---|
| antibody | FITC-conjugated anti-β tubulin antibody (Mouse monoclonal) | ProteinTech | Cat#: CL488-66240; RRID: AB_2883292 | (1:100) dilution |
| antibody | anti-*Drosophila* dynein heavy chain antibody (Mouse monoclonal) | Developmental Studies Hybridoma Bank (DSHB) | 2C11-2; RRID: AB_2091523 | (1:50) dilution |
| antibody | anti-Myc antibody (Mouse monoclonal) | doi: 10.1126/science.1061086 (*Karcher et al., 2001*) | Purified from Hybridoma cell line MYC 1-9E10.2 (RRID:CVCL_G671) | (1:100) dilution |
| antibody | anti-BicD antibody (Mouse monoclonal) | Developmental Studies Hybridoma Bank (DSHB) | anti-Bicaudal-D 4C2; RRID: AB_528103 | (1:5) dilution |
| antibody | anti-phospho-myosin light chain 2 (Ser19) (Rabbit polyclonal) | Cell Signaling | Cat# 3671; RRID: AB_330248 | (1:100) dilution |
| antibody | Fluorescein (FITC) affiniPure anti-Mouse IgG (H + L) (Goat polyclonal) | Jackson ImmunoResearch | Cat# 115-095-062; RRID: AB_2338594 | 10 µg/ml |
| antibody | Rhodamine (TRITC) affiniPure anti-Mouse IgG (H + L) (Goat polyclonal) | Jackson ImmunoResearch | Cat# 115-025-003; RRID: AB_2338478 | 10 µg/ml |
| antibody | Fluorescein (FITC) affiniPure anti-Rabbit IgG (H + L) (Goat polyclonal) | Jackson ImmunoResearch | Cat#111-095-003; RRID: AB_2337972 | 10 µg/ml |
| antibody | HRP affiniPure anti-mouse IgG (H + L) (Goat polyclonal) | Jackson ImmunoResearch | Cat# 115-035-003; RRID: AB_10015289 | 10 µg/ml |
| recombinant DNA reagent (*Drosophila melanogaster*) | pWalium22-Dlic3′UTR-shRNA | Generated in this study (available upon request, contact vgelfand@northwestern.edu) | Targeting *Dlic* 3′UTR 401–421 nt, 5′-AGAAATTTAAC AAAAAAAAAA –3′ | DNA plasmid for generating transgenic flies |
| recombinant DNA reagent (*Pyrococcus furiosus*) | pUASp-GEM | Generated in this study (available upon request, contact vgelfand@northwestern.edu) | Subcloned from pCDNA3.1-pCMV-PfV-GS-Sapphire (Addgene plasmid # 116933; RRID:Addgene_116933) | DNA plasmid for generating transgenic flies |
| recombinant DNA reagent (*Drosophila melanogaster*) | pUASp-F-Tractin-Myc-BicD | Generated in this study (available upon request, contact vgelfand@northwestern.edu) | Includes the actin binding domain of rat Inositol trisphosphate 3-kinase A (ITPKA residues 9–40), Myc and *Drosophila* BicD | DNA plasmid for generating transgenic flies |
| recombinant DNA reagent (*Physcomitrella patens*) | pUASp-F-Tractin-GFP-Kin14VIb | Generated in this study (available upon request, contact vgelfand@northwestern.edu) | Includes the actin binding domain of rat Inositol trisphosphate 3-kinase A (ITPKA residues 9–40), GFP and *Physcomitrella patens* kinesin14VIb | DNA plasmid for generating transgenic flies |
| recombinant DNA reagent (*Drosophila melanogaster*) | pUASp-Myc-HA-DlicNT-LOV2-attB | Generated in this study (available upon request, contact vgelfand@northwestern.edu) | Contains *Drosophila* Dlic residues 1–370 and LOVTRAP probe LOV2[WT] | DNA plasmid for generating transgenic flies |
| recombinant DNA reagent (*Drosophila melanogaster*) | pUASp-Zdk1-DlicCT-sfGFP-Myc-attB | Generated in this study (available upon request, contact vgelfand@northwestern.edu) | Contains LOVTRAP probe Zdk1 and *Drosophila* Dlic residues 371–493 | DNA plasmid for generating transgenic flies |
| chemical compound, drug | Rhodamine-labeled phalloidin | Thermo Fisher Scientific | Cat# R415 | 0.2 µg/ml |
| chemical compound, drug | DAPI | Sigma-Aldrich (MilliporeSigma) | D9542-1MG | 1 µg/ml |
| chemical compound, drug | 16% Paraformaldehyde Aqueous Solution, EM Grade | Fisher Scientific (Electron Microscopy Science) | Cat# 50-980-487 (EMS 15710) | 4% or 8% for fixation |

