## [Editor Report]

In their manuscript, Lu et al., use a combination of experimental approaches to determine how cellular components are transported from nurse cells into the growing oocyte during *Drosophila* egg development. The authors demonstrate that the minus-end directed microtubule motor, dynein, generates cortical flow by gliding microtubules along the cell cortex. This action is distinct from dynein's cargo transport functions, as the authors replace dynein with a minus-end directed kinesin linked to the cortex and observe the same phenomenon. This flow is capable of propelling cargoes through the ring canals into the growing oocyte via a bulk cytoplasmic transport mechanism, highlighting a novel mode of fast cytoplasmic transport. Overall, this work had broad significance to cell biologists and developmental biologists.

---

## [Decision Letter]

**Decision letter after peer review:**

Thank you for submitting your article "A novel mechanism of bulk cytoplasmic transport by cortical dynein in *Drosophila* ovary" for consideration by *eLife*. Your article has been reviewed by 3 peer reviewers, and the evaluation has been overseen by a Reviewing Editor and Anna Akhmanova as the Senior Editor. The reviewers have opted to remain anonymous.

The authors all agreed that this is a robust, innovative and thorough study that warrants publication in *eLife* once it has been revised with additional analyses and textual changes. While these are the major revisions required for publication, please address the comments on textual edits (e.g. introducing and using the terms advection or streaming for consistency within the field), as well as the comments on additional citations and discussion points raised by each reviewer.

Essential revisions:

1) Can the authors determine at which stage dynein-driven microtubule flow begins (stage 9 or earlier?) and whether this correlates with an increase in oocyte growth rate? Please see Reviewer 1's comments for additional details.

2) Can the authors analyze their current data sets to provide velocities for the movement of mitochondria (or Golgi since they appear to move at similar rates), GEMs, and microtubules? Please see Reviewer 1 and 3's comments for additional details.

3) All three reviewers noted that the last section about how Dlic is anchored to the cell cortex isn't well incorporated into the paper. The authors may want to consider how to best wrap up this section, perhaps by expanding upon this section with additional discussion about the possible linkers by citing different papers and listing the most likely candidates. Although additional experiments are not necessary for publication, we wanted to highlight this concern for the authors.

*Reviewer #1 (Recommendations for the authors):*

While one can always think of new experiments that might strengthen the paper (like acute pharmacological depolymerization of microtubules), my focus is on the rigor of some of the analysis and claims.

Throughout the paper the claim is made that this movement of microtubules and associated cytoplasm is robust in stage 9 and maybe 8 but no data are presented regarding this timing. How early does it start? It seems important to establish whether or not there really is a slow selective stage first. If so, one would expect to see a correlation with oocyte growth rates that is consistent with these the two phases. Does that exist? (Could be roughly calculated from classical data.)

Transport of objects by cytoplasmic flow is called advection. Likewise, cytoplasmic streaming is a common term for this process. It's very odd that these terms are not introduced and/or used. It's not a new concept, just new and important in this context.

Movement of mitochondria, cytoplasm, GEMs, and microtubules is variously describe as synchronized, moving together, etc. These terms seem unclear and potentially misleading. Of course, they are during the same stage and most almost certainly causally linked. Can you report movement rates? It looks like you should be able to measure them in your kymographs. One would expect that transported objects would move somewhat slower than the source of work – not necessarily measurably slower but definitely not faster.

The last section about Dlic isn't well incorporated into the paper. It's a low note to end on after a really nice paper. Maybe it could be combined with the observation that BicD is the only dynein adaptor that has (any) phenotype, which is important and gets a bit lost? Along these lines, are BicD, Spindly, and Hook all equally well expressed in oocytes? And where is BicD localized in detergent extracted egg chambers?

In the discussion about flow direction, orientation of microtubules isn't discussed, which seems crucial. Please include thoughts on this.

*Reviewer #2 (Recommendations for the authors):*

As shown by the authors, the chimeric Kinesin motor is able to restore oocyte growth and Orb localization in Dlic depleted egg chambers. Given this result, it would be interesting to determine to what extent specific cargoes can be correctly localized in this background. For cargoes such as bicoid, nanos and oskar mRNA, their localization is thought to involve both specific motor-based transport and diffusion/entrapment mechanisms. In this chimeric Kinesin background, only diffusion/entrapment mechanisms would be active. Thus, this tool might be able to provide some answers to long-standing questions in the field.

The last part of the paper that attempts to identify how Dlic is anchored at the cell cortex appears somewhat disconnected from the rest. I appreciate the effort that the authors have made to show that the anchor is not BicD or Lis1. However, as a consequence, the paper has a rather abrupt ending. In addition, related to this point, the authors suggest that a potential anchor might be Bazooka. However, Doerflinger et al., 2010 (Development) characterized a mutant in Baz that no longer localizes to the cortex. This mutant is not associated with an oocyte growth defect, suggesting that Bazooka might not be the linker.

Related to this point, I wondered why the authors did not consider the possibility that Shot might be the possible linker. In their previous manuscript, they show that depletion of Shot is also associated with a similar oocyte growth defect. Given the known cortical localization of Shot and its role as an actin-microtubule crosslinker, it might make sense to consider Shot as the potential linker.

---

## [Author Response]

Essential revisions:1) Can the authors determine at which stage dynein-driven microtubule flow begins (stage 9 or earlier?) and whether this correlates with an increase in oocyte growth rate? Please see Reviewer 1's comments for additional details.

In the revised manuscript, we specifically stated the stages of dynein-driven microtubule flows in the nurse cells (starting stage 6) and in nurse cell-to-oocyte ring canals (mostly at stage 9) (see Results p.7, lines 124-128 and Discussion p.18, lines 368-371). We also provided further discussion on the initiation of microtubule flow from the nurse cells to the oocyte (see Discussion p.18, lines 371-386) as well as updating Figure 2 -Video 2 (comparing microtubules in the ring canals at stage 7 versus stage 9) to support our Discussion section. Furthermore, we quantified the oocyte sizes from stage 4 to stage 10A and we plotted these results either as average oocyte size for each stage (Figure 2I) and the range of oocyte sizes for each stage (Figure 2—figure supplement 1B-1C). These data clearly show that the stage 9 oocyte has the highest growth rate, which is in an agreement of timing of the microtubule-driven flow to the oocyte (See Results p.9, lines 177-179).

2) Can the authors analyze their current data sets to provide velocities for the movement of mitochondria (or Golgi since they appear to move at similar rates), GEMs, and microtubules? Please see Reviewer 1 and 3's comments for additional details.

As Reviewers suggested, we analyzed the velocities of microtubules, small particles seen with bright-field microscopy, and mitochondria moving through the nurse cell-oocyte ring canals at stage 9. These new measurements are now included in Figure 2H and results (see Results p.7, lines 125-128 and p.9, lines 173-174; see Materials and methods, p.30, lines 632-638) (see Response to Reviewer #1 Point #3, pp.5-6). We also have added the velocity calculation based on oocyte size measurement in stage 9, which is consistent with our experimental data (see Discussion pp.17-18, lines 358-367) (please see more details in Response to Reviewer #1 Point #1, p.4).

3) All three reviewers noted that the last section about how Dlic is anchored to the cell cortex isn't well incorporated into the paper. The authors may want to consider how to best wrap up this section, perhaps by expanding upon this section with additional discussion about the possible linkers by citing different papers and listing the most likely candidates. Although additional experiments are not necessary for publication, we wanted to highlight this concern for the authors.

We completely agree with this comment. To better wrap up the whole story, we made following changes on the Dlic section:

1) We have now included the Dlic NT and CT data into the main figure 5 (as Reviewer #3 suggested) and included more data showing that DlicNT or DlicCT truncations alone do not rescue *Dlic-RNAi* (Figure 5C; Results p.13, lines 259-263).

2) We tested the role of the actin-microtubule crosslinker Shot in DlicCT localization as suggested by Reviewer #2, and we found that DlicCT cortical localization is independent of Shot (see Results p.14, lines 283-288; Figure 5—figure supplement 1E).

3) In the results part, we modified the Dlic conclusion to better end this part of the results and leave an open question on how Dlic gets recruited to the cortex (see Results p.14, lines 289-297).

4) In the discussion, we added several other potential linkers, including PIP5Kinase Skittles, 14-3-3 epsilon and zeta, and NuMA/Mud, in addition to Baz/Par-3. We also addressed the lack of oocyte defects in single mutants (as Reviewer #2 pointed out) and discussed the possibility of genetic redundancy of these proteins to anchor dynein to the cortex (see Discussion p.20, lines 412-428).

Reviewer #1 (Recommendations for the authors):While one can always think of new experiments that might strengthen the paper (like acute pharmacological depolymerization of microtubules), my focus is on the rigor of some of the analysis and claims.Throughout the paper the claim is made that this movement of microtubules and associated cytoplasm is robust in stage 9 and maybe 8 but no data are presented regarding this timing. How early does it start? It seems important to establish whether or not there really is a slow selective stage first. If so, one would expect to see a correlation with oocyte growth rates that is consistent with these the two phases. Does that exist? (Could be roughly calculated from classical data.)

We have included the information on the onset of microtubule movement within nurse cells (starting stage 6) and microtubule movement through nurse cell-oocyte ring canals (late stage 8-stage 9) (see Results p.7, lines 124-128 and Discussion p.18, lines 368-371). The initiation of the microtubule flow through the ring canal may be triggered by microtubule reorganization occurred in stage 7-8 oocytes as we see very different microtubule patterns in the ring canals connecting the nurse cells and the oocyte in stage 7 and in stage 9 (now we added the new data to Figure 2 -Video 2) (see Discussion p.18, lines 371-386) (more details in Response to Reviewer #1 Point #5, p.8).

We also quantified the oocyte sizes from stage 4 to stage 10A, and these data are included in Figure 2I (average size ± 95% confidence intervals for each stage) and Figure 2—figure supplement 1B-1C (individual data points for stage 9, and the ratio of max/min and 95 percentile/5% percentile for each stage). Stage 9 clearly has the highest growth rate, consistent with the timing of onset of microtubule-driven cytoplasmic advection to the oocyte.

As the Reviewer suggested, we roughly calculated the flow velocity at stage 9. During stage 9, the oocyte grows up to 10X in size within 6 hours^1^ (Figure 2-—figure supplement 1B-1C)^1^. We took the smallest oocyte size of stage 9 as the beginning size (~1400 µm^2^) and the largest oocyte size of stage 9 as the ending size (~14000 µm^2)^. To achieve such a rapid growth, each of the four ring canals connecting nurse cells to the oocyte with a diameter of ~ 6.5 µm is estimated to have a consistent flow of ~150 nm/sec throughout the whole time. This is consistent with our experimental results of the velocity of both microtubules and small spherical particles in bright-filed images (Figure 2H).

We did a similar calculation for stage 7 egg chamber and found out that to achieve the stage 7 oocyte growth (6 hours^1^ with 4 ring canals of ~5.5 µm diameter), the flow is estimated to be around 1 nm/sec. Again, this is consistent with the results that we don’t have seen apparent flow at stage 7.

Regarding the slow selective transport phase, we do believe that it does exist for following reasons:

(1) At early stages, microtubules are nucleating from the oocyte (-) to the nurse cells (+) and forms an almost “static” microtubule plug in the nurse cell-oocyte ring canals (see the last part of the newly updated Figure 2 -Video 2). It is very unlikely that such microtubule organization can support the massive flow of cytoplasm through the ring canals.

(2) AT early stages, microtubules in the nurse cell-oocyte ring canals have ~90% plus-ends towards the nurse cells, which is consistent with the cargo transport directionality driven by dynein (~90% towards the oocyte; ~10% towards the nurse cells), but is inconsistent with the direction of microtubule gliding driven by cortically-attached dynein. This 10% cargoes going towards the nurse cells indeed argues against the possibility of flow at these stages.

(3) Our gliding-only motor chimeric motor (F-Tractin-Kin14) does not fully rescue the oocyte growth; instead, the oocyte size is ~30% of the wildtype. This leads us to believe that the slow selective transport indeed contributes to the oocyte growth (see Discussion p.19, lines 389-393).

(4) Starting stage 7 to early stage 8, the microtubule system in the oocyte undergoes dramatic reorganization in order to build an anterior-posterior microtubule gradient and thus to achieve proper body axis determination. We think that it is not the best option to use fast cytoplasmic advection non-selectively carrying all cytoplasmic contents to the oocyte during the microtubule reorganization phase. This is supported by the fact that in the stage 9 oocyte partially rescued by gliding-only F-Tractin-Kin14VIb, the axis determination is disrupted (see more details in the Response to Reviewer #2 point #1, pp.12-13).

Transport of objects by cytoplasmic flow is called advection. Likewise, cytoplasmic streaming is a common term for this process. It's very odd that these terms are not introduced and/or used. It's not a new concept, just new and important in this context.

The reviewer is absolutely correct. Cytoplasmic movement is not new concept and has been studied for decades. We have replaced the term of “cytoplasmic flow” with “cytoplasmic advection” throughout the manuscript. We did not use the term of “cytoplasmic streaming”, because this is commonly referring to the bulk cytoplasmic rotation in the oocyte in stage 10B-11 (AKA ooplasmic streaming). We decided to keep the word “microtubule flow” because it describes microtubule synchronized movement, which is the cause of the cytoplasmic advection.

Movement of mitochondria, cytoplasm, GEMs, and microtubules is variously describe as synchronized, moving together, etc. These terms seem unclear and potentially misleading. Of course, they are during the same stage and most almost certainly causally linked. Can you report movement rates? It looks like you should be able to measure them in your kymographs. One would expect that transported objects would move somewhat slower than the source of work – not necessarily measurably slower but definitely not faster.

Now we have quantified the velocities of microtubules, small particles, and mitochondria in nurse cell-oocyte ring canals of stage 9 egg chambers, and the data have been included as Figure 2H, and in Results (see Results p.7, lines 125-128 and p.9, lines 173-174). As Reviewer #1 predicted, the large elongated mitochondria display a slightly slower velocity compared to microtubules, while the small particles seem to have a similar speed as microtubules (see Discussion pp.17-18, lines 362-367).

The last section about Dlic isn't well incorporated into the paper. It's a low note to end on after a really nice paper. Maybe it could be combined with the observation that BicD is the only dynein adaptor that has (any) phenotype, which is important and gets a bit lost? Along these lines, are BicD, Spindly, and Hook all equally well expressed in oocytes? And where is BicD localized in detergent extracted egg chambers?

As the reviewer suggested, we have added more results in the section of Dlic results (see Results p.13, lines 259-263 and p.14, lines 283-288) as well as expanded Dlic discussion by citing more papers on potential linkers (see Discussion p.20, lines 412-428). See more details in Essential revisions #3, pp.2-3.

BicD, Spindly and Hook are all expressed in the ovary, according to the FlyAtlas Anatomical Expression Data on Flybase (see Author response image 1). The mRNA abundance in the ovary appears to be BicD (436.6)>Hook (204.6)>Spindly (75.5). It is in an agreement with the result that BicD is the major activator for Dynein during oogenesis.

**Author response image 1. sa2fig1:** 

We examined BicD antibody staining in non-extracted and extracted samples, and the BicD staining is consistent with DHC antibody and Dlic-GFP: in extracted samples, BicD is localized to the nurse cell cortex (see Results p.11, lines 210-211, and Figure 4-—figure supplement 1E-1F).

In the discussion about flow direction, orientation of microtubules isn't discussed, which seems crucial. Please include thoughts on this.

We have now included a Discussion section about the microtubule orientation (see Discussion p.18, lines 371-386). Briefly, as shown in Author response image 2, the microtubules in the ring canals at early stages (stage 6-7) form a microtubule “plug” with the directionality that does not favor microtubule flow from the nurse cells to the oocyte. During the major microtubule reorganization at stage 7~8, the microtubule plug is “dissolved”, which allows moving microtubules in nurse cells to go through the ring canals to the oocyte at stage 9.

Reviewer #2 (Recommendations for the authors):As shown by the authors, the chimeric Kinesin motor is able to restore oocyte growth and Orb localization in Dlic depleted egg chambers. Given this result, it would be interesting to determine to what extent specific cargoes can be correctly localized in this background. For cargoes such as bicoid, nanos and oskar mRNA, their localization is thought to involve both specific motor-based transport and diffusion/entrapment mechanisms. In this chimeric Kinesin background, only diffusion/entrapment mechanisms would be active. Thus, this tool might be able to provide some answers to long-standing questions in the field.

The oocyte rescue by the chimeric kin14VIb is partial, which means the oocyte size is smaller than the control and there are various associated defects can be detected. In Figure 4 -Video 1 and in Author response image 3 of a stage 9 egg chamber (based on follicle cells and migrating border cells), oocyte nucleus is clearly mispositioned. This could be either because that individual motor-driven cargo transport is not rescued, or the anterior-posterior microtubule gradient is not properly established. As Reviewer #2 suggested, we stained the samples with Staufen, as a proxy of the posterior determinant *osk* mRNA, whose localization is known to be dependent on kinesin-1 in the oocyte. We found that Staufen was concentrated in the oocyte, which is probably carried via the cytoplasmic advection from the nurse cells to the oocyte. However, Staufen fails to localize to the posterior pole in the oocyte with the presence of endogenous kinesin-1. This lack of posterior accumulation of Staufen made us speculate that the partially rescued oocytes do not form the correct anterior-posterior microtubule gradient. The defect in microtubule reorganization could be due to the odd oocyte shape in early oogenesis (see Figure 4J, compared to Figure 4H). In Kin14 rescued samples, we very consistently observed the elongated oocyte shape, compared to more spherical-shaped oocyte in control. This abnormal shape would interfere proper signaling between oocyte and the follicle cells (for example, the posterior polar cells), and therefore disrupt the polarization events during mid-oogenesis.

**Author response image 3. sa2fig3:** 

The last part of the paper that attempts to identify how Dlic is anchored at the cell cortex appears somewhat disconnected from the rest. I appreciate the effort that the authors have made to show that the anchor is not BicD or Lis1. However, as a consequence, the paper has a rather abrupt ending. In addition, related to this point, the authors suggest that a potential anchor might be Bazooka. However, Doerflinger et al., 2010 (Development) characterized a mutant in Baz that no longer localizes to the cortex. This mutant is not associated with an oocyte growth defect, suggesting that Bazooka might not be the linker.

We have modified the Dlic section (see details in Essential Revisions #3, pp.2-3). Particularly, we have added more potential linkers for Dlic cortical anchorage and acknowledged that the displacement of Baz from the cortex does not phenocopy the dynein mutants or cause apparent oocyte growth defects. This could be explained by genetic redundancy with other linkers or a total novel mechanism independent of Baz (see Discussion p.20, lines 412-428).

Related to this point, I wondered why the authors did not consider the possibility that Shot might be the possible linker. In their previous manuscript, they show that depletion of Shot is also associated with a similar oocyte growth defect. Given the known cortical localization of Shot and its role as an actin-microtubule crosslinker, it might make sense to consider Shot as the potential linker.

Following the reviewer’s suggestion, we tested the possibility of Shot as the potential linker of Dlic. We used the strongest Shot loss-of-function mutant (*shot-RNAi* in *shot^∆EGC^* heterozygote background), which gave rise to ~100% small oocyte phenotype in our previous Current Biology paper^14^, and found that DlicCT is still localized to the nurse cell cortex. It indicates that Shot is dispensable for the cortical localization of DlicCT. We have included this set of data in Figure 5-—figure supplement 1E and in Results (p.14, lines 283-288).

References:

1 Jia, D., Xu, Q., Xie, Q., Mio, W. and Deng, W. M. Automatic stage identification of *Drosophila* egg chamber based on DAPI images. *Scientific reports* 6, 18850, doi:10.1038/srep18850 (2016).

2 Saxton, W. M. Microtubules, motors, and mRNA localization mechanisms: watching fluorescent messages move. *Cell* 107, 707-710, doi:10.1016/s0092-8674(01)00602-x (2001).

3 Bastock, R. and St Johnston, D. *Drosophila* oogenesis. *Current biology : CB* 18, R1082-1087, doi:10.1016/j.cub.2008.09.011 (2008).

4 Becalska, A. N. and Gavis, E. R. Lighting up mRNA localization in *Drosophila* oogenesis. *Development* 136, 2493-2503, doi:10.1242/dev.032391 (2009).

5 Hinnant, T. D., Merkle, J. A. and Ables, E. T. Coordinating Proliferation, Polarity, and Cell Fate in the *Drosophila* Female Germline. *Front Cell Dev Biol* 8, 19, doi:10.3389/fcell.2020.00019 (2020).

6 Lu, W., Winding, M., Lakonishok, M., Wildonger, J. and Gelfand, V. I. Microtubule-microtubule sliding by kinesin-1 is essential for normal cytoplasmic streaming in *Drosophila* oocytes. Proceedings of the National Academy of Sciences of the United States of America 113, E4995-5004, doi:10.1073/pnas.1522424113 (2016).

7 Lu, W. et al. Ooplasmic flow cooperates with transport and anchorage in *Drosophila* oocyte posterior determination. The Journal of cell biology 217, 3497-3511, doi:10.1083/jcb.201709174 (2018).

8 Jolly, A. L. et al. Kinesin-1 heavy chain mediates microtubule sliding to drive changes in cell shape. Proceedings of the National Academy of Sciences of the United States of America 107, 12151-12156, doi:10.1073/pnas.1004736107 (2010).

9 Lu, W., Fox, P., Lakonishok, M., Davidson, M. W. and Gelfand, V. I. Initial Neurite Outgrowth in *Drosophila* Neurons Is Driven by Kinesin-Powered Microtubule Sliding. Current biology : CB 23, 1018-1023, doi:10.1016/j.cub.2013.04.050 (2013).

10 Lu, W., Lakonishok, M. and Gelfand, V. I. Kinesin-1-powered microtubule sliding initiates axonal regeneration in *Drosophila* cultured neurons. Molecular biology of the cell 26, 1296-1307, doi:10.1091/mbc.E14-10-1423 (2015).

11 Winding, M., Kelliher, M. T., Lu, W., Wildonger, J. and Gelfand, V. I. Role of kinesin-1-based microtubule sliding in *Drosophila* nervous system development. Proceedings of the National Academy of Sciences of the United States of America 113, E4985-4994, doi:10.1073/pnas.1522416113 (2016).

12 Goldstein, R. E. and van de Meent, J. W. A physical perspective on cytoplasmic streaming. Interface Focus 5, 20150030, doi:10.1098/rsfs.2015.0030 (2015).

13 Doerflinger, H. et al. Bazooka is required for polarisation of the *Drosophila* anterior-posterior axis. Development 137, 1765-1773, doi:10.1242/dev.045807 (2010).

14 Lu, W., Lakonishok, M. and Gelfand, V. I. Gatekeeper function for Short stop at the ring canals of the *Drosophila* ovary. Current biology : CB, doi:10.1016/j.cub.2021.05.010 (2021).